# Novel mechanism for tubular injury in nephropathic cystinosis

**Swastika Sur[1], Maggie Kerwin[1], Silvia Pineda[1,2], Poonam Sansanwal[1], Tara K Sigdel[1], Marina Sirota[2], Minnie M Sarwal[1]***

[1]Division of Multi-Organ Transplantation, Department of Surgery, University of California, San Francisco, San Francisco, United States; [2]Bakar Computational Health Sciences Institute, University of California, San Francisco, San Francisco, United States

## eLife Assessment

This **important** study addresses the idea that defective lysosomal clearance might be causal to renal dysfunction in cystinosis. With mostly **solid** data, the authors observe that restoring expression of vATPase subunits and treatment with Astaxanthin ameliorate mitochondrial function in a model of renal epithelial cells, opening opportunities for translational application to humans.

**Abstract** Understanding the unique susceptibility of the human kidney to pH dysfunction and injury in cystinosis is paramount to developing new therapies to preserve renal function. Renal proximal tubular epithelial cells (RPTECs) and fibroblasts isolated from patients with cystinosis were transcriptionally profiled. Lysosomal fractionation, immunoblotting, confocal microscopy, intracellular pH, TEM, and mitochondrial stress test were performed for validation. CRISPR, $CTNS^{-/-}$ RPTECs were generated. Alterations in cell stress, pH, autophagic turnover, and mitochondrial energetics highlighted key changes in the V-ATPases in patient-derived and $CTNS^{-/-}$ RPTECs. ATP6V0A1 was significantly downregulated in cystinosis and highly co-regulated with loss of $CTNS$. Correction of ATP6V0A1 rescued cell stress and mitochondrial function. Treatment of $CTNS^{-/-}$ RPTECs with antioxidants ATX induced ATP6V0A1 expression and improved autophagosome turnover and mitochondrial integrity. Our exploratory transcriptional and in vitro cellular and functional studies confirm that loss of Cystinosin in RPTECs, results in a reduction in ATP6V0A1 expression, with changes in intracellular pH, mitochondrial integrity, mitochondrial function, and autophagosome-lysosome clearance. The novel findings are ATP6V0A1's role in cystinosis-associated renal pathology and among other antioxidants, ATX specifically upregulated ATP6V0A1, improved autophagosome turnover or reduced autophagy and mitochondrial integrity. This is a pilot study highlighting a novel mechanism of tubular injury in cystinosis.

## Introduction

Lysosomal storage diseases (LSD) form a significant subgroup of inherited metabolic disorders (*Anikster et al., 1999*; *Fuller et al., 2006*; *Touchman et al., 2000*; *Burton, 1998*; *Ballabio and Gieselmann, 2009*), with an incidence of more than 1:5000 live births (*Gholami Yarahmadi et al., 2022*). Cystinosis is a rare autosomal recessive LSD, caused by mutations in the *CTNS* gene, encoding lysosomal membrane transporter Cystinosin (*Kalatzis et al., 2001*). Deletion of 57 kb of the *CTNS* gene is the most common mutation that accounts for approximately 75% of the affected alleles in northern Europe (*Town et al., 1998*; *Attard et al., 1999*). A deficiency of Cystinosin results in lysosomal cystine accumulation and cystine crystal formation in virtually all tissues and organs (*Kalatzis et al.,*

*For correspondence:
minnie.sarwal@ucsf.edu

Competing interest: The authors declare that no competing interests exist.

*2001*). The kidney is the first organ affected functionally despite lysosomal cystine loading in multiple tissues and organ systems. Based on severity and age at onset of renal injury, cystinosis is phenotypically classified into nephropathic/infantile (OMIM219800; *Town et al., 1998*), juvenile (OMIM 219900; *Attard et al., 1999*), and ocular (OMIM 219750; *Anikster et al., 2000*). The most severe form is nephropathic/infantile cystinosis, characterized by the development of renal Fanconi syndrome and glomerular dysfunction, resulting in end-stage renal disease (ESRD) by 10 years of age, extendable to the second decade of life with cystine depletion therapy (*Cherqui and Courtoy, 2017*; *Elmonem et al., 2016*). Nevertheless, persistent Fanconi and progressive renal failure remains a reality for these patients despite robust adherence to cystine depletion therapies (*Wilmer et al., 2011*). These observations suggest that Cystinosin or other key molecular perturbations impact renal tubular integrity (*Bellomo et al., 2021*; *Hollywood et al., 2020*; *Jamalpoor et al., 2021*; *De Leo et al., 2020*) and function besides cystine transport (*David et al., 2019*).

Although associative functional roles for Cystinosin have been identified, including inactive mammalian target of rapamycin (mTOR) signaling (*Ivanova et al., 2016*; *Nevo et al., 2017*; *Sansanwal and Sarwal, 2012*; *Sansanwal and Sarwal, 2010*), defective autophagy, flawed clearance of damaged mitochondria by disrupted mitophagy (*Zhang et al., 2020*; *Luciani et al., 2018*; *Johnson et al., 2013*), lysosomal biogenesis (*Sansanwal and Sarwal, 2010*), abnormal tight junction-associated signaling (*Sansanwal et al., 2010*), vesicular transport defects, and increased endoplasmic reticulum (ER) stress (*Luciani et al., 2018*), the underlying drivers for these injuries have not been well identified, resulting in a paucity of new therapies to improve outcomes in nephropathic cystinosis. Furthermore, we and others have shown that renal biopsies from patients with nephropathic cystinosis reveal morphologically abnormal mitochondria and mitophagy with autophagic vacuoles, reduced mitochondrial numbers (*Sansanwal et al., 2010*; *Raggi et al., 2014*), increased ROS production (*Sansanwal et al., 2015*) and reduced lysosomal enzyme activity (*Sansanwal et al., 2010*).

We hypothesized that defective lysosomal clearance may be a pivotal cause of the altered function of renal proximal tubular epithelial cells (RPTECs). Given lysosomes ubiquitous presence in all tissues and organ-systems, we hypothesized that a comparison of transcriptional profiles of renal and extra-renal tissues from patients with nephropathic cystinosis would shed light on the unique lysosomal changes in the human kidney responsible for both the renal tubular injury in Fanconi and the progressive renal tubular injury resulting in ESRD. With this in mind, we investigated whole-genome expression profiles in paired RPTECs (*Le et al., 2020*; *Naguib, 2000*; isolated from patient urine samples) and fibroblasts (*Ambati et al., 2014*; isolated from the same patient skin biopsies) from patients with cystinosis and healthy age- and gender-matched controls. We identified dysregulation of the vacuolar (V)-ATPase multigene family, specifically ATP6V0A1, only in cystinotic RPTECs and not in normal RPTECs and fibroblasts or cystinotic fibroblasts. To further study the structural and functional consequences of ATP6V0A1 deficiency, a CRISPR/Cas9-mediated *CTNS*-knock-out (*CTNS*$^{-/-}$) immortalized RPTEC line was generated as an in-vitro model that mimicked structural and functional changes seen in patient-derived RPTECs. The *CTNS*$^{-/-}$ RPTECs also demonstrated ATP6V0A1 downregulation, resulting in significant functional (loss of intracellular pH, decreased autophagic flux, compromised mitochondrial ATP-production) and structural consequences (increased vacuolization, lack of mitochondrial cristae, reduced number of mitochondria, ER-stress, and intracellular lipid droplet (LD) formation). Further, correction of ATP6V0A1 in *CTNS*$^{-/-}$ RPTECs and treatment with antioxidants specifically, astaxanthin (ATX) increased the production of cellular ATP6V0A1, identified from a custom FDA-drug database generated by our group, partially rescued the nephropathic RPTEC phenotype (*Petri and Lundebye, 2007*). ATX is a xanthophyll carotenoid occurring in a wide variety of organisms. ATX is reported to have the highest known antioxidant activity (*Guo et al., 2015*) and has proven to have various anti-inflammatory, anti-tumoral, immunomodulatory, anti-cancer, and cytoprotective activities both in vivo and in vitro (*David et al., 2019*; *Ambati et al., 2014*; *Petri and Lundebye, 2007*; *Guo et al., 2015*; *Guo et al., 2021*; *Qiu et al., 2015*; *Ben-Nun et al., 1993*; *Foreman and Benson, 1990*). We are the first to show that ATX can induce the expression of ATP6V0A1 and has a protective effect against cystinosis-induced autophagosome formation and mitochondrial dysfunction. We have used other antioxidants, such as, Cysteamine and Vitamin E, but both had no effect on ATP6V0A1 protein expression. Our findings lead us to believe that ATP6V0A1 is not only a potential target for therapy in cystinosis, but that ATX has the potential to be repurposed as a ATP6V0A1-inducer and has potential to be used as a combination treatment with cysteamine for the renal pathology in nephropathic

cystinosis that persists despite current cystine depletion therapies (*Bellomo et al., 2021*; *Hollywood et al., 2020*; *Jamalpoor et al., 2021*).

## Results

*Figure 1A* summarizes the study design. Briefly, the study is divided into three steps. First, transcriptional study with genome microarray in RPTE and fibroblast cells. Second, protein study and functional assays in primary and lab generated RPTECs. Third, correcting the error by using plasmid or inducers to express ATP6V0A1 and checking its effect on the downstream markers. *Figure 1B* represents the schematic overview of the established (black) and hypothesized (blue) mechanisms in normal and cystinotic RPTECs. In summary, we show that compared to healthy, cystinotic RPTECs have dysfunctional *CTNS* that affects the expression of v-ATPases and SLCs, which reduces mTORC1 activity, increases autophagy, compromises mitochondrial function, and causes defective autophago-lysosomal clearance.

## Global transcriptional changes in cystinotic RPTEC affect lysosomal and mitochondrial pathways

Performed genome wide transcriptional profiling of normal and cystinotic RPTECs and skin fibroblasts with and without cystine dimethylester (CDME), used to load lysosomes with cystine to mimic the basic defect in cystinosis (*Gahl et al., 2007*; *Reimand et al., 2019*; *Figure 2*). The genetic anomaly in cystinosis (mutation of the cystine: proton transporter) is quite heterogeneous with more than 140 identified mutations (*David et al., 2019*). This heterogeneity affects the phenotype of the disease quite significantly. With this in mind all eight patients from whom the cells were obtained had the same 57 kb mutation (*Wagner et al., 2004*). Normal CDME loaded RPTECs, and fibroblasts did not transcriptionally mimic the cystinotic phenotype suggesting that CDME loading is a poor surrogate (*Sumayao et al., 2013*) in vitro model to study cystinosis tissue injury despite much of earlier research in cystinosis using CDME loading (*Figure 2*). Specific transcriptional signatures are observed in cystinotic skin-fibroblasts and RPTECs obtained from the same individual with cystinosis versus their healthy counterparts (*Figure 2B and C*). These differences between cell types, at the transcriptional level highlight tissue-specific changes in cystinosis (*Figure 2—figure supplement 1*). Nevertheless, some overlapping genes are significantly dysregulated in both cystinotic RPTECs (n=1926; FDR <0.05) and fibroblasts (n=745; FDR <0.05), with 219 overlapping genes associated with DNA integrity loss and damage. Further analysis identified that certain molecular pathways were highly enriched (*Hennings et al., 2012*) only in the kidney, and 11 significant pathways were found to be unique to cystinotic RPTECs alone (*Table 1*). Metabolic pathways, oxidative phosphorylation and acid secretion pathways are some of the significantly affected pathways in cystinosis-RPTECs. Using K-mean clustering on the genes in these significantly enriched pathways, we identified two distinct clusters (data not shown). One, is enriched in nucleus-encoded mitochondrial genes crucial for energy production, and the other is enriched in v-ATPases family, which are crucial for lysosomes and kidney tubular acid secretion. Ten lysosomal v-ATPases (*Table 2*) were downregulated in cystinotic RPTECs, five of which are significantly downregulated and some of which play important roles in proximal tubule (PT) $H^+$ secretion to support reabsorption of luminal $HCO_3^-$ and apical endocytosis (*Zhang et al., 2017*; *Zhang et al., 2019*; *Vaisbich et al., 2011*). The most significantly perturbed member of the V-ATPase gene family that was found to be downregulated in cystinosis RPTECs is ATP6V0A1 (*Table 2*), hence further attention was focused on characterization of the role of this particular gene in a human in vitro model of cystinosis.

## CRISPR-Cas9-mediated immortalized *CTNS*[-/-] RPTECs

We and others have found that CDME loading is a poor surrogate in vitro model; hence there is a need to develop a more robust in vitro model to study the renal tubular injury in cystinosis. Over time, it is likely that the primary cell lines generated from cystinotic patient urine samples may have undergone transformation and the development of a stable, robust in vitro model of renal proximal tubular epithelial cell (RPTEC) injury was recognized as an unmet need in the study of nephropathic cystinosis. Hence, we generated and validated a CRISPR/cas9-based *CTNS* gene knockout model of RPTEC injury. Ribonucleoprotein (RNP)-mediated CRISPR genome editing (*Figure 3A*) was adapted to design a guide RNA that binds to exon 3 of the *CTNS* gene in human immortalized (HuIm)

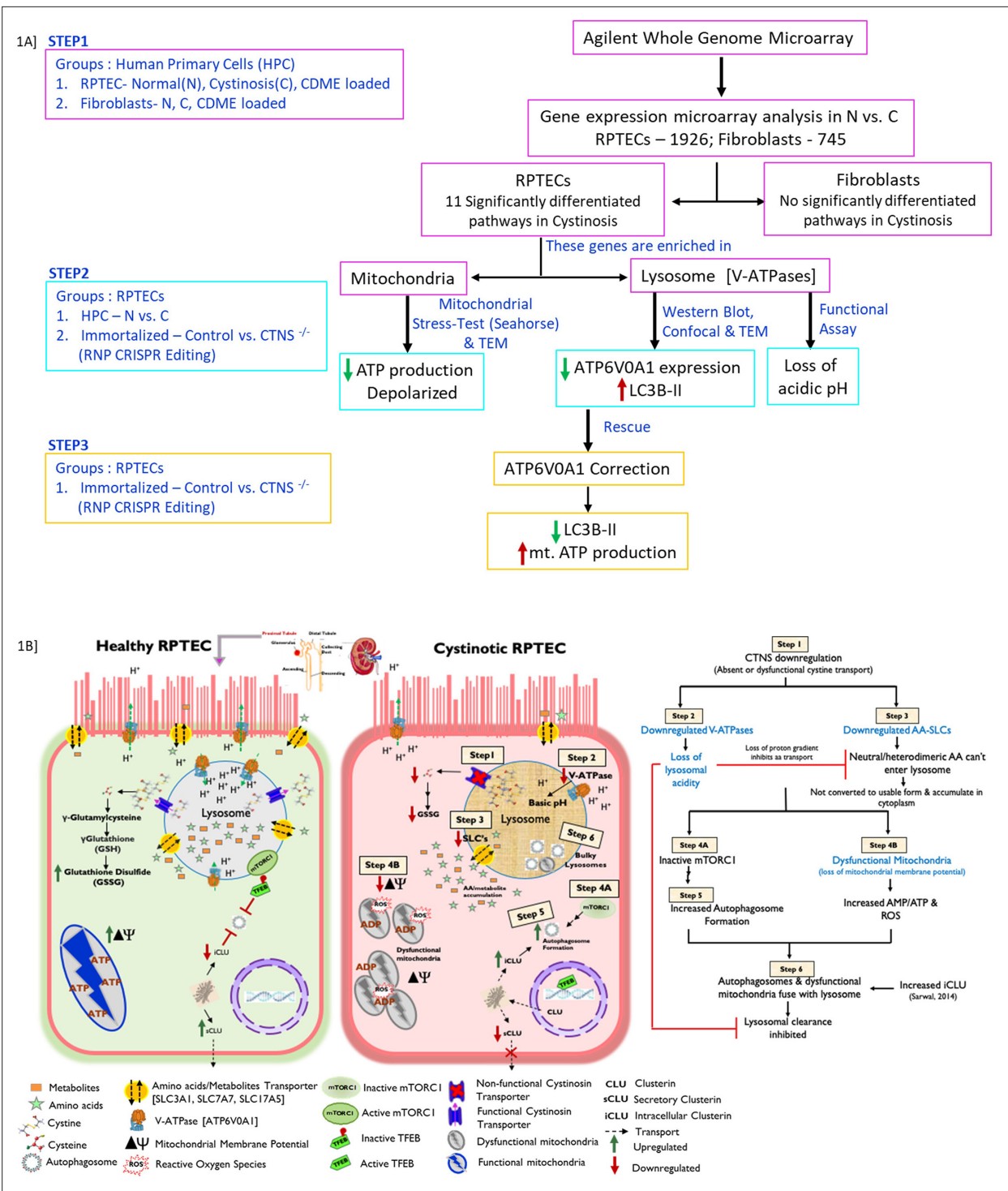

**Figure 1.** Overall study design, workflow, and schematic overview. (**A**) Transcriptome profiling with gene microarray performed on cystinotic, normal, and CDME-treated RPTECs and fibroblasts (*Step 1*). Bioinformatic analysis allowed the identification of genes that are differentially regulated in cystinosis versus normal in both the cell types. Pathway analysis was performed with all the significantly regulated genes, and 11 pathways were found to be significantly affected in cystinotic RPTECs. Most of these pathways and the genes in these pathways are crucial for lysosomal acidic pH and mitochondrial ATP production. Dysfunctional mitochondria and compromised intracellular pH were validated in *Step 2*. Our CRISPR-mediated *CTNS*⁻/⁻ immortalized renal cell line mimicked cystinosis patient primary cells isolated from their urine. Further, we rescued (*Step 3*) the disease phenotype of the cell by over-expressing ATP6V0A1, the most significantly downregulated gene among other V-ATPases. RNP, Ribonucleoprotein; V-ATPase, vacuolar ATPase; RPTEC, renal proximal tubular epithelial cells. (**B**) A schematic overview of normal and cystinotic RPTECs. We previously showed that loss of function of Cystinosin in cystinotic cells inactivates the mTORC pathway, and induces autophagy, mitophagy, and clusterin protein expression. In this

*Figure 1 continued on next page*

*Figure 1 continued*

study, we show that downregulation of *CTNS* gene also downregulates V-ATPase expression resulting in the loss of lysosomal acidity (pH). The basic pH thus blocks lysosomal clearance after autophagosome-lysosome fusion. Inhibited autophagy flux may explain why cystinotic cells have increased LC3B-II expression. Disrupted intracellular pH also impairs proton-dependent import of amino acids and other metabolites (noted in our microarray data) into the lysosome lumen, and inhibits conversion of these imported large heterodimeric amino acids into usable form. This increases the presence of metabolites in the cytoplasm that may compromise mitochondrial function and increase ROS production through an unspecified mechanism.

RPTECs and successfully knockout the gene (*CTNS*[-/-]; *Figure 3B*) with 94% efficiency. We isolated the genomic DNA, PCR amplified the *CTNS*-region with suitable primers, ran the amplified PCR product in a 2% gel to confirm the primer set that worked best, and finally submitted the amplified DNA with primers to QuintaraBio (SF, CA) for Sanger DNA sequencing. The chromatogram obtained after Sanger-seq was analyzed with TIDE webtool online (*Figure 3A*). As shown in *Figure 3B*, TIDE analysis identified an estimated percentage of insertions or deletions (indels) in the *CTNS* gene and showed that the efficiency of the *CTNS* knockout is 94% in the knockout RPTECs compared to the control RPTECs. To confirm the *CTNS* knockout at the mRNA level, we isolated total RNA, prepared cDNA, and performed qPCR (*Figure 3C*). We designed two primers targeting different *CTNS* exons – 2–3 (CTNS#1) and 9–10 (CTNS#2). Since our CRSPR-guide was targeted on exon 3 of the *CTNS* gene, we still could observe some *CTNS* mRNA expression with primer #1; however, *CTNS* expression was undetectable with primer #2. Further, at the functional level, we performed HPLC-MS/MS and showed a significant increase in cystine accumulation in *CTNS*[-/-] RPTECs (*Figure 3D*), similar to intracellular cystine accumulation found in cystinosis patients (5–6 nmol/mg protein).

## Knock out of Cystinosin in RPTECs downregulated ATP6V0A1 expression

Compared to controls, ATP6V0A1 protein expression was significantly decreased in both primary cystinotic and *CTNS*[-/-] RPTECs with an eightfold reduction in isolated lysosomal fractions (*Figure 4A–C*). Lysosomes were isolated from control and *CTNS*[-/-] RPTECs, and the purity of the isolation is shown by the presence of lysosomal marker LAMP2 and absence of GAPDH, expressed in cytoplasm (*Figure 4C*). Confocal imaging showed reduced immunopositivity for both LAMP2 (lysosomal marker) and ATP6V0A1 in cystinotic RPTECs (*Figure 4D*, *Figure 4—figure supplement 1*), with a clear expression of both in normal lysosomes. *Figure 4E–F* shows a significant reduction in intracellular acidic pH in cystinotic (pH = 5 vs normal RPTEC pH of 4.2) and *CTNS*[-/-] RPTECs (pH = 6.6 vs control human immortalized RPTECs pH of 5.8).

## Reduced autophagosome turnover and mTORC1 activity in cystinotic RPTECs

By confocal microscopy, we showed increased immunopositivity to LC3B puncta (green) and active or phosphorylated p70S6K (red) in cystinotic RPTECs compared to the control (*Figure 5A and B*). Similar to the previous figure, confocal imaging revealed decreased LAMP2 (red) immunopositivity in cystinotic RPTECs (*Figure 5A*). We performed immunoblotting to quantitatively show that both primary and *CTNS*[-/-] RPTECs, have reduced phosphorylated and total p70S6 kinase protein expression in the cystinotic RPTECs than its control (*Figure 5C and D*).

## Compromised mitochondrial function in cystinotic RPTECs

The cell mitochondria-stress test with Agilent Seahorse XFe instrument assessed mitochondrial (mt) function in *CTNS*[-/-] and cystinotic RPTECs. Drug additions, as noted in the figure, were used to assess mitochondrial bioenergetic function by measuring the oxygen consumption rate (OCR) in live cells in real time. Both cystinotic and *CTNS*[-/-] RPTECs exhibited similar patterns of compromised mitochondrial function (*Figure 6*), with significantly decreased basal, maximal and ATP-linked respiration compared to their respective controls. OCR linked to proton leak, cells spare respiratory capacity and non-mitochondrial oxygen consumption was significantly lost in cystinotic cells. There was a substantial increase in OCR linked to proton leak in normal cells, indicating that normal cells, when injected with mt-electron chain inhibitors, have higher proton leak than the diseased cells with inhibitors.

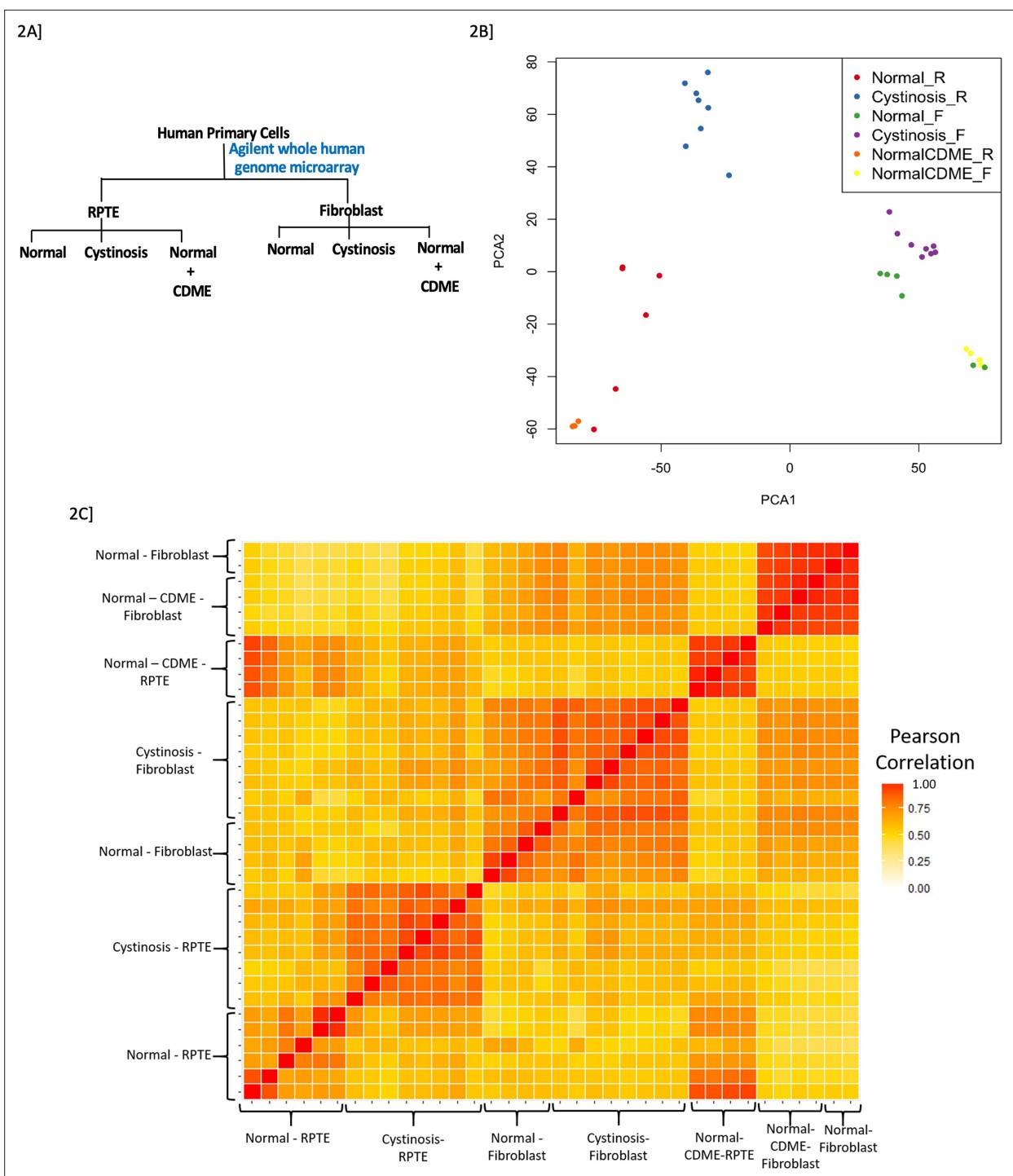

**Figure 2.** Microarray study design and transcriptional changes for cystinotic, normal, and CDM-treated RPTECs or fibroblasts. (**A**) Study design: Two types of cells were used- RPTECs and fibroblasts. For each cell type there were three groups – normal that serves as control – commercially obtained (n=6), cystinotic – obtained from individuals with cystinosis (n=8), and CDME-treated normal cells (n=4). (**B**) PCA plot further highlights the unique injury mediated gene profile in cystinotic RPTEC versus normal. Gene expression patterns of cystinotic RPTECs and fibroblasts were found to have no commonality, with cystinostic fibroblasts similar to normal. Gene expression patterns in cystinotic RPTECs are distinct and negatively correlated with normal RPTECs. Whereas gene expression in normal RPTECs laden with CDME are similar to the normal rather than disease phenotype, and hence have a positive correlation with each other. (**C**) Microarray data from both cell types are represented as a correlation heatmap. Cystinotic RPTECs showed markedly different expression profiles from both the normal and CDME loaded normal RPTEC, which was more similar to normal than diseased. Gene expression patterns in normal, cystinotic, and CDME-treated fibroblasts were found to be similar to each other depicting that nephropathic cystinosis-related changes are highly specific to renal cells.

*Figure 2 continued on next page*

*Figure 2 continued*

The online version of this article includes the following figure supplement(s) for figure 2:

**Figure supplement 1.** Amino acids and sialic acid transporters.

## Correction of ATP6V0A1 expression improved autophagosome turnover and mitochondrial function

Since most genes crucial for mitochondrial ATP-generation were downregulated in cystinotic RPTEC, we hypothesized that even without functional Cystinosin, correcting ATP6V0A1 expression would positively affect mitochondrial function and autophagosome turnover. We expressed ATP6V0A1via a plasmid in *CTNS*[-/-] RPTECs and observed reduced LC3-II accumulation (*Figure 7A*) indicating either reduced autophagy or increased autophagosome turnover. LC3-II expression was determined by dividing its expression by that of LC3-I (upper band). Beta-tubulin was used as a housekeeping gene. In addition, study of mitochondrial function (seahorse) in real time revealed increased mitochondrial basal, maximal and ATP-linked respiration - indicating an increased energy demand. ATP6V0A1 also improved non-mitochondrial oxygen consumption. However, the correction had no effect on the proton leak and significantly reduced the spare respiratory capacity (*Figure 7B*). Additionally, we showed that correcting ATP6V0A1 expression with the plasmid expressing the gene in *CTNS*[-/-] cell lines did not have any significant effect on intracellular cystine level (*Figure 7C*). Overall, ATP6V0A1 correction in *CTNS*[-/-] cells partially improved mitochondrial function compared to the *CTNS*[-/-] RPTECs transfected with control plasmid.

## Correction of ATP6V0A1 rescued morphologic renal tubular alterations

We performed TEM to characterize cellular morphologic aberrations in *CTNS*[-/-] compared to normal RPTECs and if correcting the ATP6V0A1 expression in *CTNS*[-/-] rescued the diseased RPTEC phenotype (*Figure 7D*). We observed a significant increase in autophagic vacuoles (AV), decrease in mitochondrial number with few or no cristae, swollen ER and the presence of LD within *CTNS*[-/-] RPTECs. Similar to control RPTECs, ATP6V0A1 correction significantly reduced AV (*Figure 7E–G*) and increased mitochondria number with well-preserved cristae. Nevertheless, correction did not affect intracellular LD accumulation.

**Table 1.** Pathway analysis was performed with all the significantly regulated genes, and 11 pathways were found to be significantly affected in cystinotic RPTECs.

| Significantly enriched pathways | Adjusted p-value |
|---|---|
| DNA replication | 0.0000 |
| Metabolic pathways | 0.0000 |
| Oxidative phosphorylation | 0.0014 |
| Cell cycle | 0.0014 |
| Fanconi anemia pathway | 0.0014 |
| Valine, leucine and isoleucine degradation | 0.0049 |
| Collecting duct and secretion | 0.0049 |
| Aminoacyl-tRNA biosynthesis | 0.0337 |
| Pyrimidine metabolism | 0.0330 |
| Nucleotide excision repair | 0.0337 |
| Homologous recombination | 0.0337 |

**Table 2.** List of all the v-ATPases that are significantly downregulated in cystinosis RPTECs compared to normal.

| Gene name | Fold change | q-value |
|---|---|---|
| ATP6V0A1 | 0.5867 | 0.0407 |
| ATP6V1C1 | 0.5782 | 0.0000 |
| ATP6V1B2 | 0.5736 | 0.0000 |
| ATP6VIE1 | 0.5489 | 0.4074 |
| ATP6V1H | 0.4616 | 0.0000 |
| ATP6V0E1 | 0.6500 | 0.7087 |
| ATP6V1E2 | 0.6757 | 0.2956 |
| ATP6AP1 | 0.7300 | 0.4474 |
| ATP6V1A | 0.7401 | 1.4208 |
| ATP6V0D1 | 0.7987 | 1.4208 |

The online version of this article includes the following source data for table 2:

**Source data 1.** List of the 10 v-ATPases showing no significant changes in its expression in cystinosis fibroblasts, CDME-treated fibroblasts and CDME-treated RPTECs compared to their respective controls.

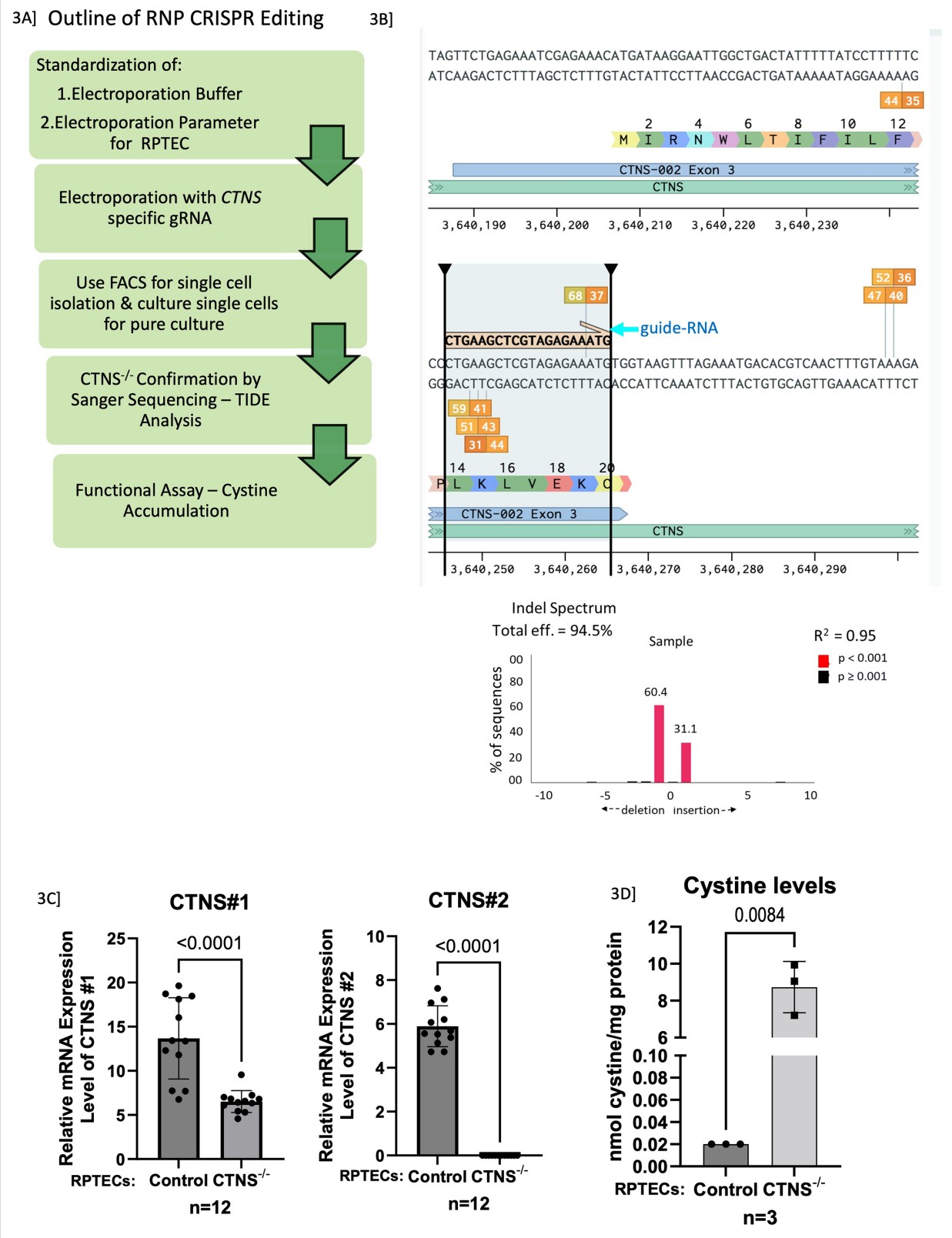

**Figure 3.** Experimental overview for the development of immortalized CTNS[-/-] RPTECs using CRISPR-Cas9 RNPs. (**A**) Outline of RNP CRISPR editing. The *CTNS* gene is located on chromosome 17p13.3 and consists of 12 exons, of which the first 2 are non-coding. Therefore, the guide RNA was targeted towards exon 3 to completely knockout the functional *CTNS* gene. CRISPR-Cas9 ribonucleoproteins (crRNPs) were synthesized in vitro to knockout *CTNS* gene and delivered to immortalize RPTECs by nucleofection. These cells were expanded for molecular validation of gene editing and

*Figure 3 continued on next page*

*Figure 3 continued*

downstream functional assays. (**B**) The guide RNA target sequence and associated PAM are highlighted on the right. We performed Sanger sequencing and the TIDE output calculating percent indels from chromatograms is depicted in the bar-graph. (**C**) Validation of *CTNS* knockout (-/-) in immortalized RPTEC by using two primers targeting different exons – *CTNS*#1 targets between exon 2–3 and *CTNS*#2 targets between exon 9–10. We have shown that in the CRISPR/Cas9 *CTNS*-/- RPTEC there is a significant reduced Cystinosin RNA levels with both the primers. (**D**) Validation of *CTNS* $^{-/-}$ in immortalized RPTEC. Increased intracellular accumulation of cystine is shown by HPLC-MS/MS method in control and *CTNS*$^{-/-}$ RPTECs. Student's t-test was used. Data are presented as mean ± SD. *p<0.05; **p<0.01; ***p<0.001.

## ATX but not other antioxidants increases ATP6V0A1 expression in *CTNS*$^{-/-}$ RPTECs

Since there is an increased production of reactive oxygen species in cystinotic renal epithelial cells (*Sansanwal and Sarwal, 2010*), we evaluated the effect of various antioxidants – Cysteamine, Vitamin E, and ATX. Both RPTECs with and without the *CTNS* were treated with 100 µM of Cysteamine for 24 hr and three different concentrations (20, 50, 100 µM) of vitamin E for 48 hr, and 20 µM of ATX for 48 hr (*Figure 8—figure supplement 1*). Both Cysteamine and Vitamin E had no significant effect on the ATP6V0A1 protein expression in *CTNS*$^{-/-}$ RPTECs (*Figure 8A and B*). Interestingly, only the ATX pretreatment increased ATP6V0A1 protein expression in *CTNS*$^{-/-}$ RPTECs (*Figure 8C*). To evaluate whether ATX could improve autophagic clearance via upregulating ATP6V0A1, LC3-II levels were measured and showed significant reduction of LC3-II/LC3-I ratio with ATX (*Figure 8C*). *Figure 8D* shows significantly increased ROS production in *CTNS*$^{-/-}$ RPTEC (p<0.0001) which was rescued with ATX treatment. MitoSox, a mitochondrial selective ROS detector, was used to measure mitochondrial-mediated ROS production and WST-1 was used to normalize cell viability. Using the JC-1 probe, we measured mt-membrane potential ($\Delta\Psi m$) since its decrease is a quintessential event in the early stages of apoptosis. Compared to control, *CTNS*$^{-/-}$ RPTEC had a significant decrease (<0.04) in the JC-1 ratio of aggregates (590 nm) to monomers (530 nm) indicating a significant loss in $\Delta\Psi m$, whereas ATX treatment rescued $\Delta\Psi m$ (*Figure 8E*) in *CTNS*$^{-/-}$ RPTEC.

## Discussion

Despite whole-body cystine loading, cystinosis predominantly affects the human kidney, causing renal Fanconi syndrome soon after birth and progressive renal damage. These processes are slowed but not evaded by cystine-depleting therapies (*Cherqui and Courtoy, 2017*; *Sansanwal et al., 2010*). Over the years, multiple studies have shown some potential for cysteamine-combined treatments, both in in vivo and in vitro models. But none are yet approved by the FDA (*Bellomo et al., 2021*; *Hollywood et al., 2020*; *Jamalpoor et al., 2021*; *De Leo et al., 2020*). In our study, transcriptional microarray analysis identified that the majority of the genes affected in cystinosis belong to one large cluster, which is crucial for normal lysosomal and mitochondrial function (*Table 1*). We identified a list of vacuolar (v)-ATPases and ATP6V0A1 (*Table 2*) as the most downregulated genes in cystinotic RPTECs, which were also found to have a role in multiple cystinosis-related dysregulated pathways in our dataset (*Tables 1 and 2*). Thus, we identified a unique disruption in lysosomal pathways of RPTECs harvested from patients with nephropathic cystinosis. This was not observed in normal RPTECs, nor in paired skin fibroblasts from the same cystinosis patients.

To assess and cross-validate the structural and functional impact of these lysosomal changes in human cystinotic RPTECs, we generated a *CTNS*$^{-/-}$ in vitro, HuIm RPTEC cystinosis model (*Jamalpoor et al., 2021*; *Hollywood et al., 2022*). We found an association between the knocking out of the *CTNS* and the downregulation of v-ATPases. We selected *ATP6V0A1*, the most downregulated v-ATPase, crucial for lysosomal acidification and investigated its association with the cystinosis phenotype. We aimed to find out if this relationship could explain the early occurrence of lysosomal acidification defect in renal Fanconi syndrome and progressive structural and functional loss of renal reserve. Cystinotic cells are known to have an increased autophagy or reduced autophagosome turnover rate (*Ivanova et al., 2016*; *Nevo et al., 2017*). Autophagic flux in a cell is typically assessed by examining the accumulation of the autophagosome or autophagy-lysosome marker LC3B-II. This accumulation can be artificially induced using bafilomycin, which targets the V-ATPase, thereby inhibiting lysosomal acidification and degradation of its contents. Taken together, the observed innate increase in LC3B-II in cystinotic RPTECs (*Figure 5A*) without bafilomycin treatment suggests dysfunctional lysosomal

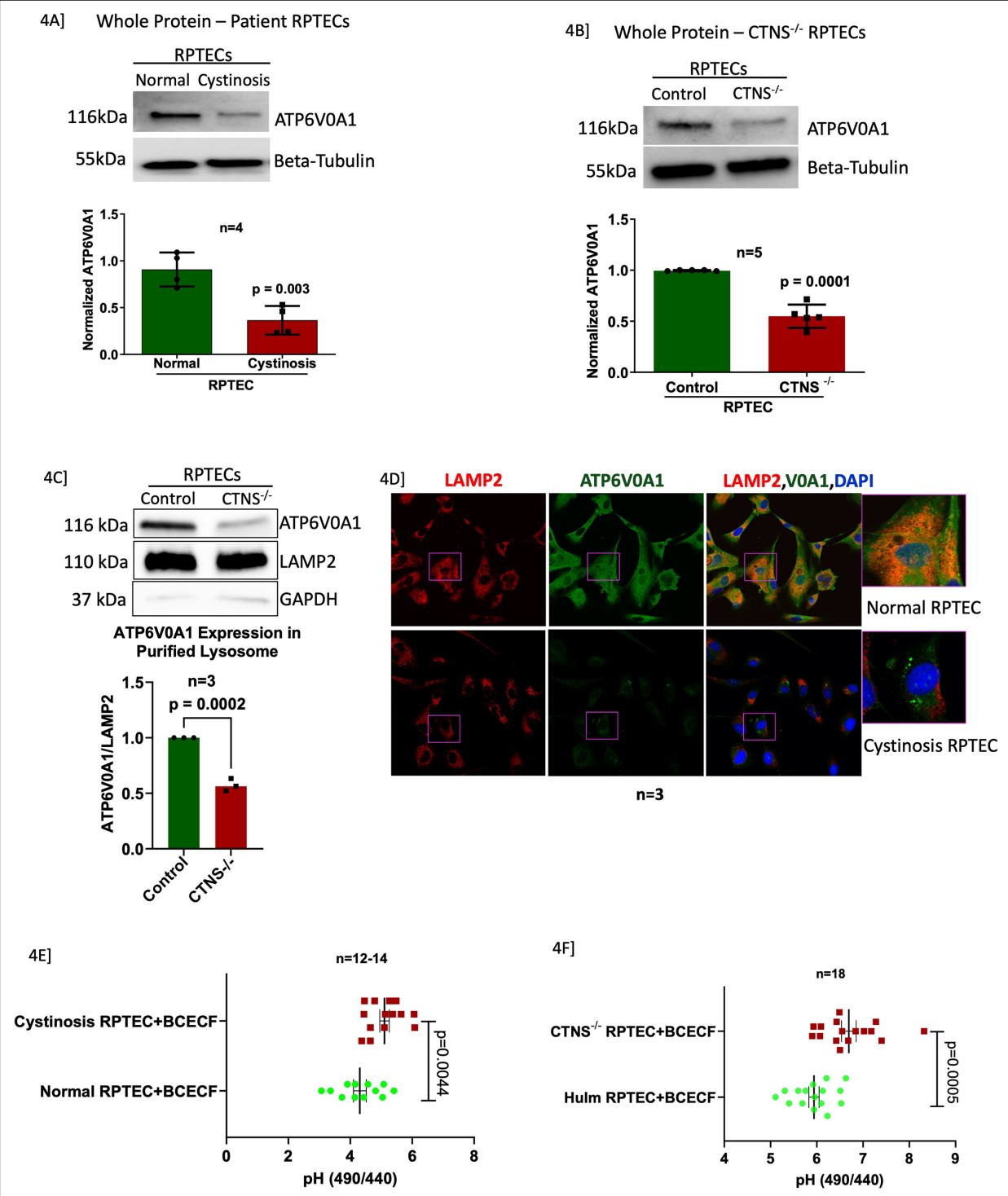

**Figure 4.** Vacuolar-ATPase, ATP6V0A1, is downregulated in both, cystinotic and *CTNS* -/- RPTECs, further confirming that our in vitro cellular model behaves like primary cystinotic renal cells. This highlights that in addition to the loss of lysosomal Cystinosin there are also other key biological changes in the lysosomes in nephropathic cystinosis. (**A-B**) Immunoblot analysis of the expression of ATP6V0A1 in cystinotic and *CTNS* -/- RPTECs with their respective controls. (**C**) Immunoblot analysis of the expression of ATP6V0A1 in lysosomal fraction isolated from control and *CTNS* -/- RPTECs. Absence of GAPDH expression depicts successful isolation of pure lysosomes from RPTECs. Results are represented as mean ± SD. The statistically significant differences between the two groups are indicated in the figure. Student's t-test (* p≤0.05; **p≤0.01; ***p≤0.001). There was no difference between LAMP2 expression in control and cystinotic RPTECs, making this marker a good control for the experiment. (D) Immunofluorescence and confocal microscopy showing endogenous LAMP2 and ATP6V0A1 distribution in normal and cystinotic RPTECs. Colocalization of ATP6V0A1 and LAMP2 is

*Figure 4 continued on next page*

*Figure 4 continued*

shown in the right panel. Results are representative of at least three separate experiments. Cystinotic and *CTNS* -/- RPTECs have an intracellular basic pH compared to their respective controls. Cells were stained with BCECF-AM and measured with a microplate fluorimeter in triplicate wells. The pH-dependent spectral shifts exhibited by BCECF allow calibration of the pH response in terms of the ratio of fluorescence intensities measured at two different excitation wavelengths (490, 440 nm). (**E**) RPTECs isolated from cystinosis patient urine had a more basic pH than normal individuals (\*\*p≤0.01). (**F**) Similarly, to the primary cells, our in vitro cellular model, *CTNS*-KO RPTEC, also had a more basic pH than control. The unpaired t-test indicated significant differences (\*\*\*p≤0.001) between control and diseased cells. Data are representative of at least three biological replicates. Data are presented as mean ± SD.

The online version of this article includes the following source data and figure supplement(s) for figure 4:

**Source data 1.** Original files for western blot analysis displayed in *Figure 4A–C*.

**Source data 2.** Tiff file containing original western blots for *Figure 4A–C*, indicating the relevant bands and treatments.

**Figure supplement 1.** Immunofluorescence and confocal microscopy showing endogenous LAMP2 and ATP6V0A1 distribution in human immortalized RPTEC and CTNS-/- RPTECs.

acidification and thus could be linked to inhibited v-ATPase activity. Similar to the study by Andrzejewska et al. (*Sansanwal and Sarwal, 2012*), we have observed reduced immunopositivity and expression of total and phosphorylated-p70S6 kinase protein, a direct downstream substrate of mTORC1 (*Figure 5B–D*) and a common marker for mTORC1 activity. This might be due to the essential role of Cystinosin in active mTORC1 signaling. Its absence results in decreased levels of both total and phosphorylated p70S6k in cells, directly impacting downstream processes like endocytosis and mTORC activity.(*Ivanova et al., 2016*; *Nevo et al., 2017*; *Sansanwal and Sarwal, 2012*; *Andrzejewska et al., 2016*; *Le et al., 2020*; *Gahl and Thoene, 2002*; *Brodin-Sartorius et al., 2012*). Further, we identified drugs with strong antioxidant activity by using a collated FDA-drug functional database (*Coffey et al., 2014*): Cysteamine, vitamin E, and ATX. However, ATX was the only agent that upregulated ATP6V0A1 and provided recovery of the specific RPTEC injury phenotype in cystinosis.

Our data indicates that the absence of functional Cystinosin downregulates ATP6V0A1 expression (*Figure 4*), resulting in loss of intracellular acidic pH, decreased autophagy flux, increased autophagosome accumulation (*Figure 7*), loss of mTORC1 activity (*Figure 5*), and a compromised mitochondrial structure and function (*Figures 6 and 7*); moreover, collectively the results suggest a possible mechanistic explanation for the PT dysfunction in cystinosis (*Figure 1B*). Interestingly, all these pathophysiology known to be compromised in cystinotic RPTECs, and are not recovered by Cysteamine (*Jefferies et al., 2008*; *Zoncu et al., 2011*).

V-ATPases play a critical role in the acidification of organelles within the endocytic, lysosomal, and secretory pathways and are required for ATP-driven proton transport across membranes (*Hughes and Gottschling, 2012*). Hence, the measurement of lysosomal pH was important. However, none of the available methods to measure lysosomal pH (*Webb et al., 2021*; *DePedro and Urayama, 2009*; *Ma et al., 2017*) worked for cystinotic RPTECs as these cells are too sensitive; therefore, we used BCECF-AM dye, which is a cell-membrane permeable dye that can enter the cell and its organelles. Intracellular pH measured using BCECF-AM represents the overall pH of the cytoplasm and its organelles. We showed that the absence of functional Cystinosin results in less acidic or more basic intracellular-organelle pH in cystinotic RPTECs (*Figure 4E and F*). An increase of approximately 0.8 pH units in cystinotic RPTECs results in a 6.3-fold difference in hydrogen ion concentration, which is significant enough to disrupt the cellular function of normal RPTECs. According to published literature, an elevation of 0.2–0.3 pH units in lysosomes has been associated with reduced pH-dependent cleavage and chronic alterations in autophagy and degradation processes. Re-acidifying the lysosomes with cAMP was shown to reverse these changes (*Coffey et al., 2014*). In both yeast and humans, the absence of an acidic pH within lysosomes impairs their degradation function, leading to lysosomes filled with autophagosomes (*Marchi et al., 2014*; *Farmer et al., 2020*) and also reducing pH-dependent amino acid storage in the vacuolar lumen, which causes mitochondrial dysfunction. (*Gahl et al., 1988*). Our work will be the first to demonstrate that in cystinotic RPTECs, the absence of Cystinosin affects v-ATPase expression, disrupts the acidic pH balance in the endolysosomal system (*Figure 4*), and impairs mitochondrial function. (*Figure 6*).

Successful transfection of a plasmid carrying the ATP6V0A1 gene into *CTNS*-/- RPTECs corrected the v-ATPase expression levels (*Figure 7A*). At the functional level, correcting ATP6V0A1 expression in *CTNS*-/- RPTECs reduced LC3-II protein expression, indicating decreased autophagy or improved

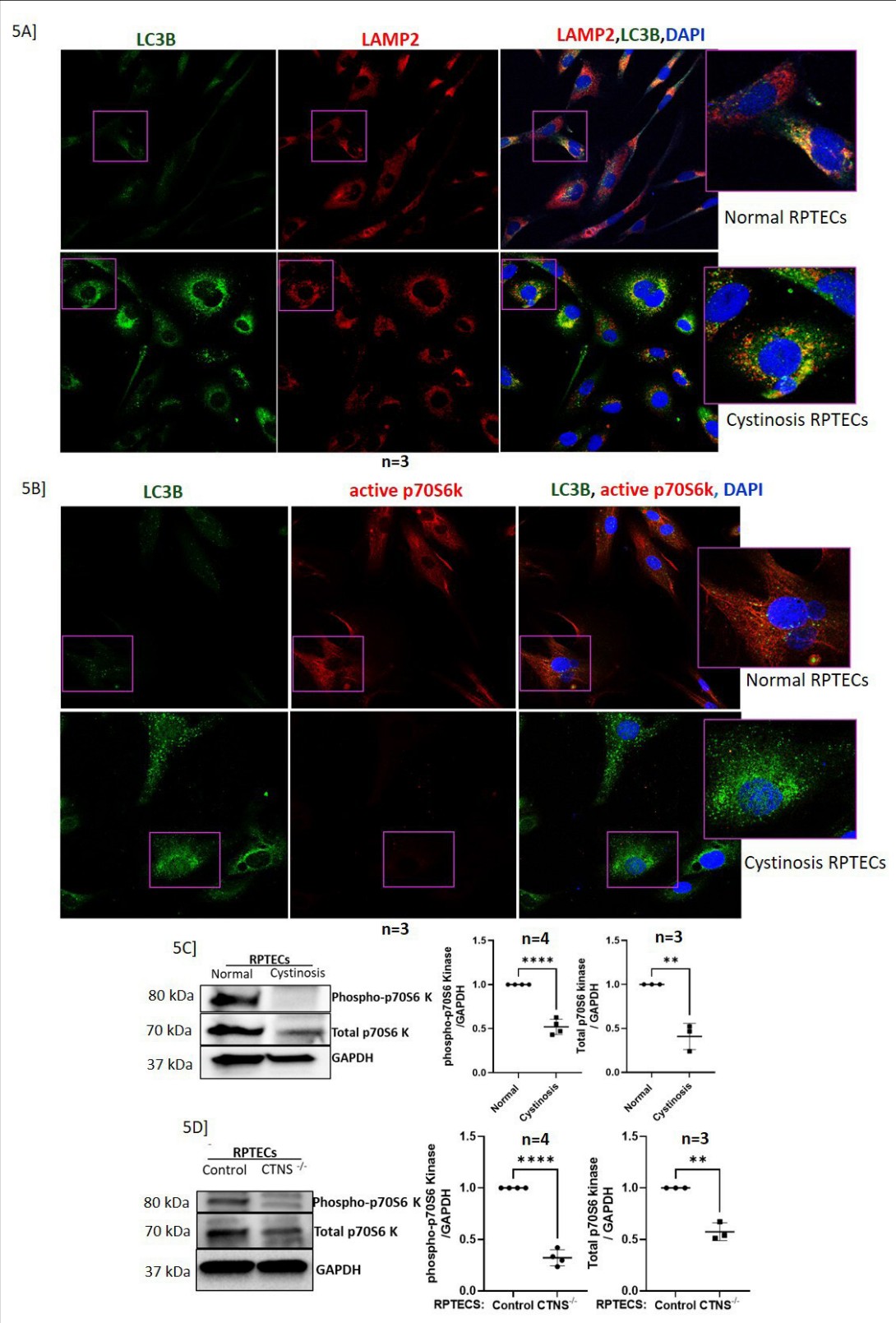

**Figure 5.** Increased immunopositivity to LC3B-II, or autophagy, and decreased immunopositivity to phospho-p70S6 kinase, or mTORC1 activity. (**A**) Immunofluorescence and confocal microscopy showing endogenous LC3B and LAMP2 distribution in normal and cystinotic RPTECs. Colocalization of LC3B and LAMP2 is shown in the right panel. (**B**) Immunofluorescence and confocal microscopy showing endogenous LC3B and phospho-p70S6 kinase (marker of mTOR activity) distribution in normal and cystinotic RPTECs. Colocalization of LC3B and phosphor-p70S6 kinase is shown in the right

*Figure 5 continued on next page*

*Figure 5 continued*

panel. (**C-D**) Immunoblot analysis of the expression of phosphorylated-p70S6 kinase and total p70S6 kinase in cystinotic and *CTNS* $^{-/-}$ RPTECs with their respective controls. Since there is a change in total p70S6K expression in with and without the *CTNS*, we normalized both, the phosphorylated and total protein, to GAPDH. mTORC1 plays a central role in cell growth, proliferation, survival, and autophagy inhibition via AMPK, therefore, the presence of lower activity of mTORC1 in cystinosis supports the observations of increased cell death and autophagy as shown by increased LC3B puncta. Results are representative of at least three separate experiments (biological replicates). Results are represented as mean ± SD. The statistically significant differences between the two groups are indicated in the figure. Student's t-test (* $p \leq 0.05$; ** $p \leq 0.01$; *** $p \leq 0.001$).

The online version of this article includes the following source data for figure 5:

**Source data 1.** Original files for western blot analysis displayed in *Figure 5C and D*.

**Source data 2.** Tiff file containing original western blots for *Figure 5C and D*, indicating the relevant bands and treatments.

autophagosome turnover (*Figure 7A*), and partially improved mitochondrial function (*Figure 7B*). However, correcting ATP6V0A1 had no effect on cellular cystine levels (*Figure 7C*), likely because Cystinosin is known to have multiple roles beyond cystine transport. Cystinosin is demonstrated to be crucial for activating mTORC1 signaling by directly interacting with v-ATPases and other mTORC1 activators. Cystine depletion using cysteamine does not affect mTORC1 signaling (*Andrzejewska et al., 2016*). Our data, along with these observations, further supports that Cystinosin has multiple functions and that its cystine transport activity is not mediated by ATP6V0A1.

Structurally, this correction increased the number of mitochondria and improved their structure (*Figure 7E–F*) and decreased the number of autophagosomes or vacuoles by either enhancing autophago-lysosomal clearance or reducing autophagy (*Figure 7D*). These findings suggest that ATP6V0A1 plays a critical role in autophagosome turnover and mitochondrial function in *CTNS* $^{-/-}$ RPTECs. However, more research is needed to fully understand how ATP6V0A1 regulates these mechanisms in *CTNS* $^{-/-}$ RPTECs.

The compromised mitochondrial function observed in cystinotic cells (*Figure 6*) suggests that the mitochondrial inner membrane in these cells is already damaged, as the injection of inhibitors had little effect on the already compromised mitochondria. The findings indicate that the decline in total ATP-linked respiration is primarily due to reduced mitochondrial ATP production. Alteration in mitochondrial functions (*Figure 6*) likely also results in ER stress or vice versa; ER stress was previously demonstrated in nephropathic cystinosis. Our study revealed increased ER wrapping of mitochondria in *CTNS* $^{-/-}$ RPTECs (*Figure 7F*), a phenomenon referred to as mitochondria-ER-associated membranes (MAMs). This observation is known to occur in ER that is stressed (*Bernardini et al., 1985*). MAMs mediate apoptosis, which is known to be higher in nephropathic cystinosis. However, curiously, ATP6V0A1 correction did not rescue ER-wrapping of mitochondria or intracellular LDs (*Figure 7G*) in *CTNS* $^{-/-}$ RPTECs. LDs are known to be induced by inflammation and ROS and have been linked with neurodegenerative disorders (*Jefferies et al., 2008*). Previously, LDs have been shown in muscle biopsies from patients with cystinosis (*Sakai et al., 2019*; *Chitchumroonchokchai et al., 2004*); however, further studies are needed to understand LD's role in renal pathology in cystinosis. V-ATPases are needed for ATP-driven proton transport across membranes (*Yang et al., 2018*), which explains the significant downregulation of several amino acid and metabolite transporters, including SLC17A1, SLC17A3, SLC17A5, SLC3A1, and SLC7A7 (*Figure 2—figure supplement 1*) in cystinotic RPTECs. These findings were a little surprising as there is no clear explanation as to why sialic acid transporters would be affected. Moreover, to gauge the clinical impact of this perturbation we would require an exclusive study.

Another notable finding of our study is the specific effect of antioxidant ATX on *CTNS* $^{-/-}$ RPTEC, which corrects the ATP6V0A1 levels, enhances autophagosomal turnover, improves cystinosis-induced mitochondrial dysfunction, and rescues mitochondrial membrane potential. The rescue pattern of ATX in *CTNS* $^{-/-}$ RPTECs is in many ways similar to the metabolic correction of *CTNS* knock out cells by plasmids bearing ATP6V0A1. This suggests that ATX may ameliorate the impaired autophagy phenotype occurring in cystinotic RPTEC via regulating ATP6V0A1 expression. However, further studies are warranted to understand how ATX regulates ATP6V0A1 expression and its other protective mechanism in *CTNS* $^{-/-}$ RPTECs. We noticed a significant increase in cystine levels with ATX treatment alone (data not shown in the manuscript), while the combined ATX and cysteamine treatment significantly reduced cystine accumulation to the normal level. This may suggest that when co-administered with cysteamine, they have the potential to complement each other's shortcomings. We believe that the

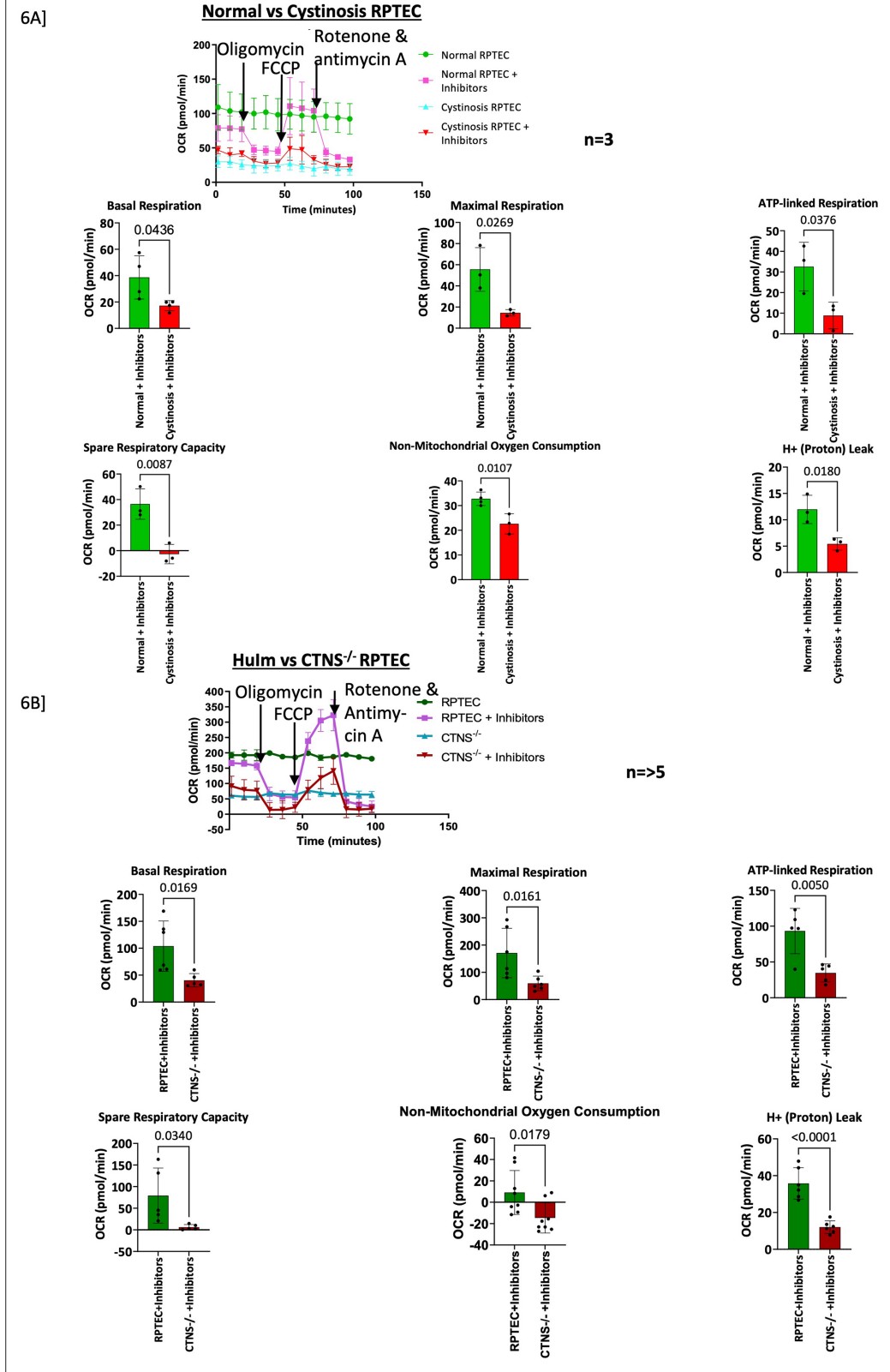

**Figure 6.** The Agilent Seahorse Mitochondria Stress Test detected mitochondrial defects in primary cystinotic and *CTNS*[-/-] RPTECs compared to their respective controls. (Top Center) XF Cell Mito Stress Test assay design and standard output parameters. (**A**) XF Cell Mito Stress Test shows that primary cystinotic RPTECs have diminished basal and maximal respiration, ATP-linked respiration, spare respiratory capacity, non-mitochondrial oxygen

*Figure 6 continued on next page*

*Figure 6 continued*

consumption, and proton leak compared with its control (normal RPTEC) when both were treated with specific electron transport chain (ETC) inhibitors. (**B**) Similar to the primary diseased cells, the *CTNS*$^{-/-}$ also demonstrated diminished mitochondrial activity. Both cystinotic and *CTNS*$^{-/-}$ RPTECs exhibited similar patterns of compromised mitochondrial function, with significantly (Normal vs. Cystinosis: p<0.025, p<0.0019, p<0.023; human immortalized RPTEC (HuIm) vs. *CTNS*$^{-/-}$: p=0.006, p=0.001, p=0.0002, respectively) decreased basal and maximal respiration and ATP-linked respiration compared to controls. The OCR linked to proton leak and cells spare respiratory capacity was significantly (Normal vs. Cystinosis: p<0.007, p<0.0007; HuIm vs. *CTNS*$^{-/-}$: p<0.0001, p=0.002 respectively) lost in cystinotic cells. Non-mitochondrial oxygen consumption did not significantly change in cystinotic or *CTNS*$^{-/-}$ RPTECs. There was a substantial (Normal vs. Cystinosis: p<0.007; HuIm vs. *CTNS*$^{-/-}$: p<0.0001) increase in OCR linked to proton leak. Student's t-test. Values represent mean ± SD for at least three independent experiments (* p≤0.05; **p≤0.01; ***p≤0.001). OCR, oxygen consumption rate.

increase in cystine with ATX alone could be due to interactions between ATX's ketone or hydroxyl groups and cystine's amine or carboxylic groups. Further research on this interaction is ongoing. In various disease models, ATX is shown to regulate mitogen-activated protein kinase (MAPK) by inhibiting JNK1/2 activation (*Wang et al., 2016*; *Qiao et al., 2017*), affect AMP-activated protein kinase (AMPK) by inhibiting mTOR pathway (*Deng et al., 2019*), affect Cerulein mediated increase in LC3 expression—causing reduced apoptosis, autophagy and improved cell potency (*Kim et al., 2010a*; *Kim et al., 2010b*; *Zhang et al., 2018*; *Yan et al., 2016*; *Ikeda et al., 2008*)— and impedes inflammation by inhibiting the JAK/STAT3 pathway (*Wang et al., 2010*). ATX is also known to ameliorate oxidative stress, ER stress, and mitochondrial dysfunction (*Lin et al., 2017*; *Bhuvaneswari et al., 2014*; *Kitahara et al., 2017*; *Kleta et al., 2004*; *Nesterova et al., 2015*; *Wilmer et al., 2007*). Cumulatively, all these studies support ATX as a promising therapeutic agent for the treatment of a wide variety of diseases. We are currently planning additional in vivo experiments to study ATX's effects on lysosomal degradation.

There are a few limitations to our study, which are inherent to any life science studies involving humans, wherein experimental controls cannot be tightly imposed. Firstly, since cysteamine treatment does not reverse Fanconi syndrome or inhibit ESRD, we hypothesized the presence of other confounding genes that could be affected in cystinosis other than *CTNS*. These would be those genes, whose expression or lack of it, cannot be corrected by cystine-depleting therapy. However, there are a few reports of preservation of some renal tubular function when cysteamine was initiated early after birth, but even these patients succumb to ESRD in their second decade (*Sansanwal et al., 2010*; *Sansanwal et al., 2015*). Typically, Cysteamine treatment is initiated at one year of age, and maybe, even at this young age, lysosomal cystine-accumulation has started, and at the cellular level irreversible tubular cell damage is progressing to renal Fanconi or ESRD. Second, we also acknowledge that our RPTECs were exposed to CDME for only 30 min, after which they began to lose cystine from their lysosomes because of their normal contingent of Cystinosin (*Wilmer et al., 2007*). Regardless of how long the cells were harvested after loading, the exposure of the cells to increased lysosomal cystine cannot be compared to longstanding exposure of the knock-out cells to lysosomal cystine. This also explains the similarity in transcriptional profiles of CDME-loaded cells to normal and cystinotic cells. Third, we also noted the loss of some epithelial markers in the patient derived RPTECs, which may have occurred with time but is less relevant to this paper as we are looking at a focused pathway in nephropathic cystinosis. We recognized the need for other cell lines that closely mimic the cystinosis-mediated renal pathology; hence, we created the CRISPR-mediated *CTNS*$^{-/-}$ RPTECs. Nevertheless, we believe that urine RPTECs harvested from Cystinosis patients and the *CTNS*$^{-/-}$ model system are closer mimics of the human disease phenotype and, hence, suit our pursuit—the understanding of molecular injury pathways in cystinotic RPTECs. Fourth, BCECF-AM measures whole cell rather than lysosomal pH, but due to the fragility of the cystinotic RPTEC, direct lysosomal pH measurement was not possible. We acknowledge that additional studies needed to understand the interplay between transcriptional regulations of ATP6V0A1 in cystinosis and if *CTNS*$^{-/-}$ cells, chronically treated with cysteamine, acquire any further changes to ATP6V0A1 expression. An important observation in our paper is that the abnormalities present in RPTECs are absent in fibroblasts; we believe that some of these differences are due to the unique changes in v-ATPases, many of which are tissue-specific in expression. A thorough study of other tissues that are also affected by cystinosis, such as the retina, esophagus, skeletal muscle, CNS, and endocrine tissues may provide a more valuable information. It

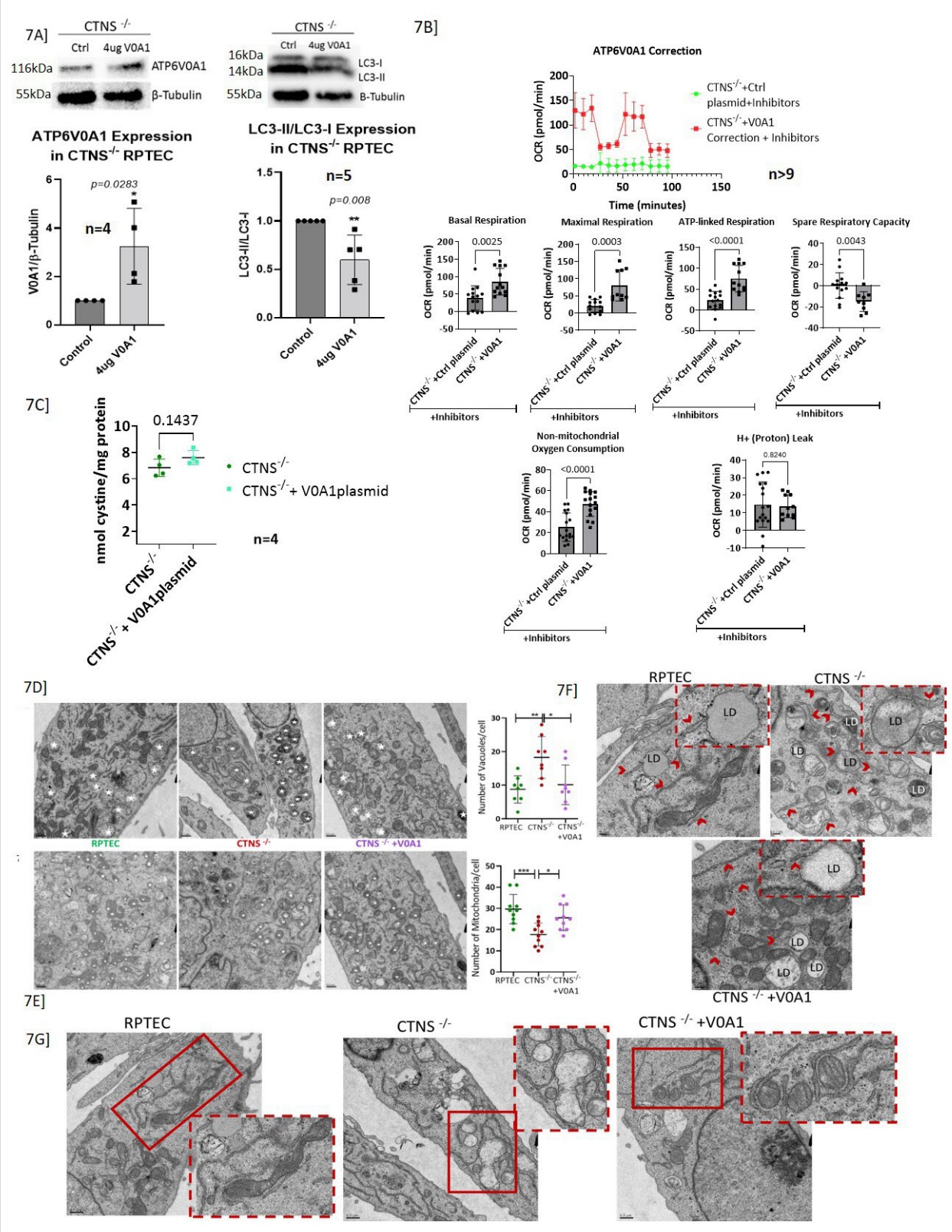

**Figure 7.** Correction of ATP6V0A1, by transfecting the *CTNS*-/- RPTECs with a Myc-DDK-tagged human ATP6V0A1-plasmid, partially rescued the disease cell phenotype. (**A**) Immunoblot showing successful correction of ATP6V0A1 protein in the *CTNS*-/- cells. Correction of ATP6V0A1 significantly reduced the LC3-II protein expression. (**B**) The XF Cell Mito Stress Test shows that correction of ATP6V0A1 in *CTNS*-/- cells significantly improved basal respiration, and increased maximal respiration and ATP-linked respiration. Overall, ATP6V0A1 correction in *CTNS*-/- cells most closely resemble the normal human

*Figure 7 continued on next page*

*Figure 7 continued*

immortalized (HuIm) RPTECs compared to *CTNS*-/- or *CTNS*-/- with control plasmid. Electron microscopic evaluation of *CTNS*-/- RPTECs, and its partial recovery by overexpressing ATP6V0A1. (**C**) Correction of ATP6V0A1 in *CTNS*-/- cells had no effect on its cystine level. (**D**) TEM of control, *CTNS*-/-, and ATP6V0A1 overexpressed *CTNS*-/- (*CTNS*-/- +V0A1) RPTECs at low magnification. Asterisks indicate autophagic vacuoles (AV). Number of AV per cell from eight different sections of each sample was counted and depicted as a bar graph. Significant increase in AV (p=0.006) is observed in *CTNS*-/- compared to that of RPTECs and *CTNS*-/- +V0A1 groups. (**E**) TEM of control, *CTNS*-/-, and *CTNS*-/- +V0A1 RPTECs at low magnification. Asterisks indicate mitochondria. Number of mitochondria per cell from eight different sections of each sample was counted and depicted as a bar graph. Significant decrease (p=0.0005) in mitochondria is observed in *CTNS*-/- compared to that of RPTECs and *CTNS*-/- +V0A1 groups. Both figures A and B indicates rescue of the diseased phenotype by the correction of ATP6V0A1 (p=0.019; p=0.02, respectively). (**F**) Structurally abnormal ER-wrapped mitochondria with lesser cristae was observed in *CTNS*-/- compared to that of RPTECs. Though, structurally normal mitochondria with cristae was observed in *CTNS*-/- +V0A1 group, but majority of the mitochondria are wrapped by ER. Scale = 0.5 µm. (**G**) Intracellular LD was observed in more number of cells in *CTNS*-/- and *CTNS*-/- +V0A1 samples than control RPTECs. LDs have a unique structure, which is delimited by a monolayer of phospholipids differing from the classical bilayer structural organization. Arrows indicate endoplasmic reticulum (ER). Scale = 0.2 um. One-way ANOVA or t-test as suitable. Values represent mean ± SD for at least three independent experiments (* p≤0.05; **p≤0.01; ***p≤0.001).

The online version of this article includes the following source data for figure 7:

**Source data 1.** Original files for western blot analysis displayed in *Figure 7A*.

**Source data 2.** Tiff file containing original western blots for *Figure 7A*, indicating the relevant bands and treatments.

is possible that minor and slower onset of functional perturbations in extra-renal organs is largely due to the intracellular cystine accumulation in lysosomes and not due to V-ATPase changes, as seen in the kidney. More specialized and targeted studies are warranted to study V-ATPase as a confounding gene in cystinosis and to confirm it as the pathophysiological variable behind the slow evolution of renal Fanconi and the progression of renal tubular and renal cortical damage, resulting in renal failure by the second decade of life. Understanding this mechanism of injury will be critical also for early detection of kidney damage, as renal injury detection at a functional level is delayed as compensatory mechanisms in the kidney result in a delayed rise in the serum creatinine, used to calculate the renal reserve by the eGFR formulae (*Bjornsson, 1979*).

In summary, the novel findings of this study are ATP6V0A1's role in cystinosis-associated renal pathology and, among other antioxidants, ATX specifically upregulated ATP6V0A1, improved autophagosome turnover, reduced autophagy, and secured mitochondrial integrity. Although this is only a pilot study, our finding that ATX, in vitro, can ameliorate cystinosis-associated dysfunctional pathways is of paramount importance and requires further study in cystinotic animal models, to clarify its utility in clinical settings.

# Methods

**Key resources table**

| Reagent type (species) or resource | Designation | Source or reference | Identifiers | Additional information |
|---|---|---|---|---|
| Cell line (*Homo-sapiens*) | RPTECs; Skin Fibroblasts | Gift from Drs. Gahl and Racusen | | Primary |
| Cell line (*Homo-sapiens*) | RPTEC | Cambrex Biosciences | | Primary |
| Cell line (*Homo-sapiens*) | Skin Fibroblasts | Coriell Cell Repositories | | Primary |
| Cell line (*Homo-sapiens*) | Immortalized RPTEC | Kerafast | Cat No. ECH001 | Immortalized cell line, Male |
| Drug | Astaxanthin (ATX) | Millipore Sigma | Cat. No SML0982 | |
| Drug | Cysteamine | Millipore Sigma | Cat. No M9768 | |
| Drug | Vitamin E | Selleckchem | Cat. No S4686 | |
| Transfected Construct (CRISPR/cas9) (*Homo-sapiens*) | guide RNA, tracrRNA, and Cas9 | Benchling/Dharmacon | | |

*Continued on next page*

*Continued*

| Reagent type (species) or resource | Designation | Source or reference | Identifiers | Additional information |
|---|---|---|---|---|
| Sequence-based reagent | qPCR primer#1 primer#2 | Thermos Fisher | #1: Assay ID: Hs01568699_m1 #2: Assay ID: Hs01568706_m1 | |
| Commercial assay, kit | Whole Human Genome 4×44 k 60-mer oligonucleotide arrays | Agilent | G4112F | Microarray-transcriptional profiles |
| Commercial assay, kit | Lysosome enrichment kit | Pierce Biotechnology | PI89839 | Lysosome Isolation |
| Antibody | ATP6V0A1 | Synaptic Systems | Cat. No. 109002 | (1:500 dilution) |
| Antibody | LAMP2 | Santa Cruz Biotechnology | Cat. No. sc18822 | (1:2000 dilution) |
| Antibody | LC3B | Cell Signaling | Cat. No. 3868 | (1:1000 dilution) |
| Antibody | phospho-p70 S6 Kinase (Thr389) | Millipore Sigma | Cat. No. MABS82 | (1:500 dilution) |
| Antibody | Total- p70 S6 Kinase | Cell Signaling | Cat. No. 9202 S | (1:1000 dilution) |
| Antibody | GAPDH | Cell Signaling | Cat. No. 97166 S | (1:2000 dilution) |
| Antibody | beta-tubulin | Cell Signaling | Cat. No. 2128 S | (1:2000 dilution) |
| Other | BCECF AM | Invitrogen | B1170 | pH measurement |
| Commercial assay, kit | Seahorse XF Cell Mito Stress test | Agilent Technologies | 103015–100 | |
| Recombinant DNA reagent | Myc-DDK-tagged ATP6V0A1 | Origene | RC226206 | Plasmid |
| Commercial assay, kit | JC-1 mitochondrial membrane potential | Abcam | ab113580 | |
| Commercial assay, kit | MitoSox Red superoxide indicator | Thermo Fisher | M36008 | |

## Study design and samples

Human RPTECs and fibroblasts were isolated from eight unique individuals with a biochemically, clinically and genetically confirmed diagnosis of nephropathic cystinosis (provided by Dr. Gahl; *Moran et al., 1990*; *Uniyal et al., 2019*); in addition, similar cell types, RPTEC (Cambrex Biosciences, East Rutherford, NJ) and fibroblast (Coriell Cell Repositories, Camden, NJ) cell lines were commercially obtained as normal control cells. The patient derived RPTECs has its limitations, which are highlighted in the discussion section. Both normal RPTECs and fibroblasts were treated with cysteine dimethyl ester (CDME) to artificially load lysosomes with cystine, as this method has been used previously as a disease model for cystine RPTEC loading, although it is unclear how accurate this model is for functional analysis of molecular changes in cystinosis patient derived RPTECs (*Elmonem et al., 2016*; *Schneider et al., 1967*). In addition, we utilized a human immortalized (HuIm) RPTEC line (Clone TH1, passage 8, Cat No. ECH001, Kerafast, Boston, MA), as control, and used CRISPR-Cas9 to generate a *CTNS*-/- knock out HuIm RPTECs that was then structurally and functionally characterized to evaluate it as an in vitro model system to study human cystinosis RPTEC injury. This purchased HuIm RPTECs are derived from primary human renal proximal tubule epithelial cells (RPTECs) immortalized by two lentiviral vectors carrying the human telomerase and the SV40 T antigen. All cells were verified to be free of mycoplasma contamination. Details of the study design and its findings are shown in *Figure 1*.

All RPTECs were cultured in renal epithelial growth medium (REGM; Lonza, Bend, OR); fibroblast cells were cultured in Minimum Essential Media (MEM) with Earl's salts, supplemented with 15% FBS, 2 mM L- glutamine, 2 X conc. of non-essential AA, 100 µg/ml Penicillin, 100 U/ml Streptomycin and 0.5 µg/ml Fungizone (Invitrogen Corporation, Carlsbad, CA) at 37 °C in a 5% $CO_2$ atmosphere. The medium was changed every alternate day, and cultured cells were harvested with 0.05% Trypsin/EDTA (Lonza, Bend, OR) and passaged. All cells were cultured in a 95% air/5% $CO_2$ Thermo Forma incubator (Thermo Fisher Scientific, Waltham, MA) at 37 °C. All the experiments with cystinotic fibroblasts and RPTECs were performed between passage numbers 2–7, and normal immortalized RPTECs purchased from the company were used even at later passages as long as the cells looked healthy under microscope. For CDME loading, cells were treated with 1 mM CDME (Sigma-Aldrich, St. Louis, MO) for 30 min. Cells were pretreated with 20 µM ATX, purchased from Millipore Sigma, Burlington, MA (Cat. No SML0982) for 48 hr. Cysteamine was purchased from Millipore Sigma, Burlington, MA (Cat. No

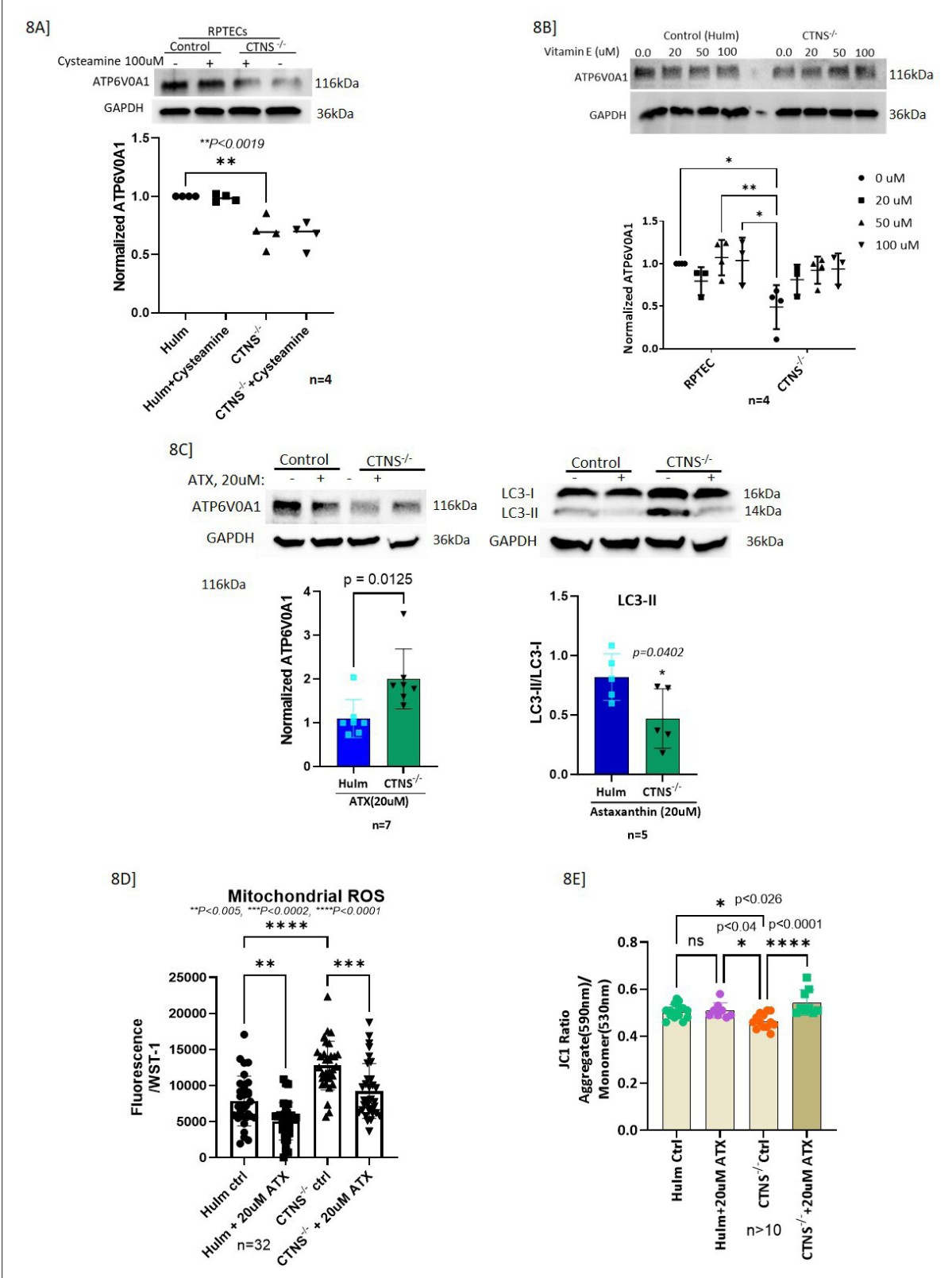

**Figure 8.** Astaxanthin (ATX) but not Cysteamine or Vitamin E upregulates ATP6V0A1 and rescued the cystinosis RPTEC phenotype. (**A**) Immunoblot showing that 100 µM of cysteamine treatment for 24 hr did not increase the expression of ATP6V0A1 in *CTNS*[-/-] RPTECs. (**B**) Immunoblots showing VitaminE has no effect on ATP6V0A1 expression in *CTNS*[-/-] RPTECs. (**C**) Immunoblots showing ATX upregulates ATP6V0A1, reduced the LC3-II (p=0.04) protein expression. Values represent mean ± SD for three independent experiments (* p≤0.05; **p≤0.01; ***p≤0.001). (**D**) ATX reduced (p=0.0002)

*Figure 8 continued on next page*

*Figure 8 continued*

mitochondrial ROS normalized to cell viability measured by WST-1. (E) ATX improved mitochondrial membrane potential (p=0.002, p<0.0001 at both 10 µM and 20 µM, respectively) in *CTNS⁻/⁻* as shown by the JC1 ratio. One-way ANOVA followed by Tukey's test or t-test as suitable. Values represent mean ± SD for atleast three independent experiments (* p≤0.05; **p≤0.01; ***p≤0.001).

The online version of this article includes the following source data and figure supplement(s) for figure 8:

**Source data 1.** Original files for western blot analysis displayed in *Figure 8A–C*.

**Source data 2.** Tiff file containing original western blots for *Figure 8A–C*, indicating the relevant bands and treatments.

**Figure supplement 1.** ATX dose response curve.

M9768) and cells were pretreated with Cysteamine for 24 hr. Vitamin E was purchased from Selleck-chem, Houston TX (Cat. No S4686) and cells were treated with it for 48 hr. For each of the treatments, the control well of cells was treated with the same solvent where the compound was dissolved in.

## Adaptation of CRISPR-Cas9 method to generate *CTNS⁻/⁻* RPTECs

Since variations in genetic background and characteristics of the patient and normal RPTECs can result in experimental variations, independent of the *CTNS* mutation, immortalized healthy RPTEC cell lines (Kerafast, Boston, MA) were used to generate isogenic *CTNS⁻/⁻* cell lines. CRISPR-Cas9 ribo-nucleoproteins (crRNPs) were synthesized in vitro by the incubation of *CTNS*-specific guide RNA, trans-activating crRNA (tracrRNA), and Cas9 protein (Dharmacon, Lafayette, CO). These preformed complexes were then delivered to immortalized RPTECs by nucleofection for editing. We used bench-ling to design the guide RNA to specifically cut at exon 3 (*Tusher et al., 2001*).Though this is not a known cystinosis-causing mutation but such a knockout causes complete inactivation of Cystinosin, which is the common output of all the known mutations associated with nephropathic cystinosis. Since efficiency of gene-knockout varies from one cell to other, we sorted single cells by FACS and created a pure culture from a single cell., amplified over the cut-site using touch-down PCR ampli-fication, and then submitted the PCR products for Sanger sequencing using both the forward and reverse TIDE oligos originally used for amplification. Once the chromatograms were returned, we chose to populate and use cells with at least 95% allelic editing, which provided a rough estimate of knock-out percentages (*Figure 3*). Briefly, we cultured these single cells and isolated the DNA from both, the control RPTECs and *CTNS⁻/⁻* RPTECs, and then performed PCR amplification over the cut site from their genomic DNA. We designed single-stranded DNA oligos to serve as PCR primers for *CTNS* over the targeted cut site. We then amplified over the cut site using a touch-down PCR ampli-fication strategy with appropriate annealing temperatures for the *CTNS*-specific primers. After this step we submitted the amplified DNA along with both the forward and reverse oligos originally used for amplification the *CTNS* gene for Sanger sequencing. If the nucleofection was successful, then it will be evident in the chromatogram obtained after Sanger. An estimated percentage of Indels was generated by uploading the experimental and control chromatograms to the TIDE webtool online.

## Validation of the generated *CTNS⁻/⁻* RPTECs at the functional level by HPLC-MS/MS

To detect the phenotype of these newly developed *CTNS⁻/⁻* cell lines, we performed a functional assay measuring intracellular cystine levels using an HPLC-MS/MS method (UCSD Biochemical Genetics, San Diego). Briefly, we prepared the sample by trypsinizing the adherent cells, washed the cell pellet with ice cold 1 mL distilled PBS, centrifuged at 500 × *g* for 5 min, resuspended the cell pellet in 150 µl ice cold 650 µg/mL N-Ethylmaleimide (NEM; Sigma-Aldrich, St. Louis, MO) in PBS solution, then performed cell-dissociation, followed by adding 50 µL of 15% Sulfosalicylic Acid (SSA; Sigma-Aldrich, St. Louis, MO), centrifuged, saved the cell pellet for protein estimation and collect the supernatant separately, bring the volume up to 0.5 ml. The samples were stored at –80 °C until transfer (on dry ice) to the UCSD Biochemical Genetics lab. The pellet was resuspended in 0.5 mL 0.1 N NaOH to the cell protein pellets to solubilize, pulse vortexed, placed the tube on a rocker with gentle agitation overnight, then protein concentration was calculated by using a standard Pierce BCA Protein Assay Kit (Thermo Fisher Scientific, Hampton, NH). The standards for the BCA assay were resuspended and diluted in 0.1 N NaOH instead of water. High levels of intracellular cystine in the generated *CTNS⁻/⁻* cell line compared to control cells confirmed the successful knockout of the *CTNS* gene. The level of

intracellular cystine accumulation in *CTNS*[-/-] cell lines were comparable to cystine levels in cystinosis patients (*Elmonem et al., 2016*; *Schneider et al., 1967*; *Figure 3C*).

## Validation of the generated *CTNS*[-/-] RPTECs at the transcript level by qPCR

We isolated total RNA from control and *CTNS*[-/-] RPTECs by using RNeasy Mini kit (Cat# 74104; QIAGEN, Hilden, Germany). We followed the standardized protocol provided by the company. The RNA was stored in –80°C for long-term storage. We used 200 µg of the RNA for the complementary-DNA (cDNA) preparation or reverse transcription. Briefly, we added VILO master mix (Thermos Fisher, Waltham, MA) that contains all the reaction components in a pre-mixed formulation and nuclease-free water to the RNA for cDNA synthesis in a thermal cycler (Eppendorf) using lab standardized cDNA synthesis method. The cDNA was stored at 4° C to be used next day for quantitative Polymerase Chain Reaction (qPCR). For qPCR, we designed two primers targeting two specific exons on *CTNS* gene – primer#1 targets between exon 2–3 and primer#2 targets between exon 9–10. These primers were connected to Taqman MGB probe (Thermo Fisher, Waltham, MA). We diluted the cDNA and used 1.25 ng for the qPCR reaction. Briefly, we added master mix (Applied Bioscience, Waltham, MA) that contains all the reaction components, nuclease-free water, and primers to the cDNA and loaded the 384 PCR plate to the thermal cycle (Applied Bioscience, Waltham, MA) using lab standardized qPCR template.  Raw Ct data normalized using the delta delta Ct method against 18 S and a human universal reference RNA was uploaded into Partek Genomics Suite v.6.6 (Partek Inc, St. Louis, MO, USA). Data were analyzed with Student's t-test to determine any statistically significant differences between groups. All data are presented as mean ±SD. All statistical analyses were performed in Partek Genomics Suite v.6.6., GraphPadPrim v.8. (GraphPad Software Inc) and in Microsoft Excel (Microsoft, USA).

## RNA isolation for microarray

Cells were grown until 70–80% confluence and processed for total RNA extraction using RNeasy Midi Kit (QIAGEN Inc, Germantown, MD). Total RNA concentration was measured by NanoDrop ND-1000 (NanoDrop Technologies, Wilmington, DE) and the integrity of the extracted total RNA was assessed with the Agilent 2100 Bioanalyzer using RNA Nano Chips (Agilent Technologies, Santa Clara, CA). Total RNA was stored at –80 °C until preparation for the microarray experiments.

## Microarray experiments to characterize RPTEC and fibroblast transcriptional profiles in nephropathic cystinosis

Hybridization of samples was conducted on Agilent Whole Human Genome 4×44 k 60-mer oligonucleotide arrays (G4112F, Agilent Technologies, Santa Clara, CA), using 150 ng of total RNA as template/sample. The arrays were scanned on an Agilent scanner and further processed using Agilent Feature Extraction Software (Agilent Technologies, Santa Clara, CA).

## Lysosomal fractionation

Lysosomal fractions were isolated from cultured cells by density gradient separation using the lysosome enrichment kit (Pierce Biotechnology, Waltham, MA) for both tissue and cultured cells, following the protocol provided by Pierce. Fraction purity was assessed by western blot using lysosome-specific antibody against LAMP2.

## Western blot

Whole cells and lysosomal extracts were prepared, and an equal amount of protein (15 µg) was subjected to SDS-PAGE. All primary antibody incubations were done in PBS supplemented with 0.1% Tween-20 (vol/vol) and 5% milk (wt/vol) overnight followed by washing with PBS-Tween (PBS supplemented with 0.1% Tween). The primary antibodies used were: ATP6V0A1 (Cat. No. 109002, Synaptic Systems, Goettingen, Germany), LAMP2 (Cat. No. sc18822, Santa Cruz Biotechnology, Santa Cruz, CA), LC3B (Cat. No. 3868, Cell Signaling, Danvers, MA), phospho-p70 S6 Kinase (Thr389) (Cat. No. MAB S82, Millipore Sigma, Burlington, MA), total p70 S6 kinase (Cat. No. 9202 S Cell Signaling, Danvers, MA), GAPDH (Cat. No. 97166 S, Cell Signaling, Danvers, MA), and beta-tubulin (Cat. No. 2128 S, Cell Signaling, Danvers, MA). The Peroxidase-conjugated secondary antibodies were diluted

1:2000 in PBS-Tween, incubated with the blot for a minimum of 1 hr at room temperature, and then washed with PBS-Tween and developed using Amersham ECL Plus Detection Reagent (RPN2124, Millipore Sigma, Burlington, MA). Loading levels were normalized using 1:2000 anti-GAPDH or beta-tubulin and anti-LAMP2 Abs. Band quantification was performed using the ImageLab software (National Institutes of Health).

## Measurement of pH

To measure lysosomal pH, we tried two methods – (1) pHLARE (*Webb et al., 2021*) and (2) LysoSensor (*Ma et al., 2017*). But due the fragility of the cystinotic cells, optimum fluorescence level of the biosensor, pHLARE could not be reached. Again, lysoSensor treatment to measure the pH killed the cystinotic RPTECs rapidly even at a very low concentration. Finally, the conversion of non-fluorescent 2',7'-bis-(2-carboxyethyl)–5-(and-6)-carboxyfluorescein acetoxymethyl ester (BCECF AM; Invitrogen, Waltham, MA) into a pH sensitive fluorescent indicator by the intracellular esterase was used to measure the pH, which represents an overall cytoplasmic and organelle pH and is not specific to the lysosomes. Briefly, we seeded a fixed number of cells in a clear-bottom black 96-well plate and incubated overnight in a $CO_2$ incubator for the cells to attach. Next day, we incubated the cells with 2 μM of BCECF for 30 min, washed the plate with Hank's Balanced Salt Solution (HBSS) then BCECF fluorescence was measured by using fluorescence microplate reader and pH was calculated. The fluorescence ratio was acquired using the SpectraMax iD3 plate reader (Molecular Devices, San Jose, CA; excitation = 490, 440 nm; emission = 535 nm). The ratio of BCECF fluorescence at 490/440 nm is a function of pH.

## Measurement of mitochondrial oxygen consumption rate

The Agilent Seahorse XFe analyzer allows for real-time measurements of cellular metabolic function in cultured cells. The oxygen consumption rate (OCR) was measured by the extracellular flux analyzer XF24 (Seahorse Bioscience, Santa Clara, CA) following optimization of cell number per well. RPTECs were plated at $4 \times 10^5$ cells/well in a Seahorse 24-well V7 microplate (Seahorse Bioscience, Santa Clara, CA) and cultured in complete renal epithelial growth medium for 16–18 hr in a 5% $CO_2$ incubator at 37 °C. Cells were counted carefully and an equivalent optimum cell density (4x105 cells/ well) was used to always seed the same number of cells on the same plate. Additionally, background correction wells (i.e. wells that have not been seeded with cells) were included in the assay to normalize the data to background plate noise. Prior to the assay, the cells were washed and incubated with assay media (Agilent Technologies, Santa Clara, CA) supplemented with 1 mM glucose (Agilent Technologies, Santa Clara, CA), 1 mM pyruvate (Agilent Technologies), and 2 mM glutamine (Agilent Technologies, Santa Clara, CA) at 37 °C without $CO_2$ for 45 min. Mitochondrial function was measured using Seahorse XF Cell Mito Stress test (Agilent Technologies, Santa Clara, CA). Mitochondrial complex inhibitors (1.5 μM of oligomycin, 0.5 μM of FCCP, 0.5 μM of rotenone and antimycin A) were freshly prepared in XF assay media prior to each experiment and were distributed in ports surrounding the sensor which were sequentially injected to each well.

OCR following serial injection of various probes was used as an indicator of mitochondrial function. Oligomycin, an ATP synthase inhibitor, was utilized as a probe for ATP-linked oxygen consumption; carbonyl cyanide-4-(trifluoromethoxy)phenylhydrazone (FCCP), an oxidative phosphorylation uncoupling agent, was used to induce maximum oxygen consumption and the resultant OCR was used to calculate spare respiratory capacity. A mixture of rotenone and antimycin-A inhibited complex I and complex III was used to result in complete inhibition of mitochondrial respiration and determination of non-mitochondrial oxygen consumption. We compared the pattern observed after injection of each inhibitor in cystinotic RPTECs.

## Plasmid-mediated ATP6V0A1 expression in *CTNS* $^{-/-}$ RPTECs

Myc-DDK-tagged ATP6V0A1 expression plasmid (RC226206, Origene, Rockville, MD) for ATP6V0A1 induction and pCMV6-Entry, mammalian vector with C-terminal Myc-DDK Tag (PS100001, Origene, Rockville, MD) as control were used. Briefly, $3 \times 10^5$ *CTNS*$^{-/-}$ and control RPTECs were seeded in six-well plates and incubated for 24 hr prior to transfection. TurboFectin 8.0 Transfection Reagent (F81001, Origene, Rockville, MD) was used at a final concentration of 4 μg/ml for transduction. The correction of ATP6V0A1 expression was verified with western blotting analysis.

## Confocal microscopy

For immunofluorescence, RPTECs were plated in 4-well Chamber Slide with removable wells (Thermo Fisher, Waltham, MA), fixed in 100% chilled methanol (5 min), permeabilized with PBS containing 0.25% Triton X-100 (10 min), and washed three times in PBS. Cells were incubated in 10% normal goat serum blocking solution (Invitrogen, Waltham, MA) for 1 hr followed by overnight primary antibody incubation in a 4 °C humidified chamber. For co-immunostaining, we added both primary antibodies raised in different host species at the same time. The next day, slides were washed three times in PBS and followed by secondary antibody incubation in 1% BSA for 1 hr at room temperature in the dark. After washing with PBS, the cells were counterstained with DAPI for 5 min and then washed. The 1.5-mm-thick coverslip was then mounted with a drop of ProLong Glass Antifade Mountant (Thermo Fisher, Waltham, MA). Primary antibodies used were: LAMP2 (Santa Cruz Biotechnology, Santa Cruz, CA), ATP6V1B2 (Abcam, Cambridge, United Kingdom), ATP6V0A1 (Synaptic Systems, Goettingen, Germany), LC3B (Cell Signaling Technology, Danvers, MA), phospho-p70S6 Kinase (Thr398; Millipore Sigma-Aldrich, St. Louis, MO). Secondary antibodies, donkey anti-Rabbit IgG Alexa Fluor 488 and goat anti-Mouse IgG Alexa Fluor 555 (Thermo Fisher, Waltham, MA) were used to detect bound primary antibody. Slides were viewed using a Leica SP5 Confocal Laser Scanning Microscope, and the images were analyzed by Leica Confocal software.

## Transmission electron microscopy (TEM)

$3 \times 10^5$ RPTECs were seeded in each well of a six well plate and maintained for 24 hr. On the day of the experiment, fresh 2% glutaraldehyde was generated from an 8% stock glutaraldehyde in complete REGM culture media and was added to each well so that cells were fully covered with fixative. After 15 min, the fixed cells were scrapped gently and transferred to a microcentrifuge tube. The samples were centrifuged, and the supernatant was discarded followed by the quick addition of 1 ml fresh fixative (2% glutaraldehyde in 0.1 M Cacodylate buffer pH 7.2). At this stage, the cells were stored at 4 °C and were then handed over to the Electron Microscope Laboratory (EML) imaging core at the University of California, Berkeley for sectioning and imaging. Sections were cut at 80 nm, stained with lead citrate and uranyl acetate, and examined under an FEI Tecnai12 electron microscope (FEI). The electron micrographs obtained from multiple distinct low-powered fields were used to count the number of mitochondria and autophagic vacuoles per cell in at least eight different view fields for each cell culture sample, and the average number of mitochondria or autophagic vacuole per cell culture was calculated.

## Mitochondrial membrane potential assay

The mitochondrial membrane potential ($\Delta\Psi$m) was measured with the JC-1 mitochondrial membrane potential assay kit (ab113580, Abcam, Cambridge, United Kingdom) according to manufacturer's instructions. Briefly, RPTEC were seeded at 12,000 cells/well and allowed to adhere overnight in a black clear-bottom 96-well plate. Cells were treated with or without 10 or 20 µM ATX for 48 hr and then washed once with 1 X dilution buffer prior to incubation with 20 µM JC-1 dye for 10 min at 37 °C. Following incubation, cells were washed twice with 1 X dilution buffer and fluorescence intensity was determined for red aggregates (excitation = 535 nm)/emission = 590 nm and green monomers (excitation = 475 nm/emission = 530 nm) with the SpectraMax iD3 plate reader (Molecular Devices, San Jose, CA). The ratio of JC-1 aggregates (590 nm) to JC-1 monomers (530 nm) was calculated. A decrease in aggregate fluorescent count is indicative of depolarization whereas an increase is indicative of hyperpolarization.

## Mitochondrial ROS production and cell viability

Mitochondrial ROS production was assessed using MitoSox Red superoxide indicator (Thermo Fisher, Waltham, MA). Briefly, cells plated at 10,000 cells/well in 96-well plates were washed with Hanks balanced salt solution (HBSS; Thermo Fisher, Waltham, MA) and treated with 5 µM MitoSOX for 15 min at 37 C and 5% $CO_2$, protected from light. After staining, cells were washed twice with HBSS to remove background fluorescence. Fluorescence was read (excitation = 510 nm, emission = 580 nm) with the SpectraMax iD3 plate reader (Molecular Devices, San Jose, CA). Following MitoSox assay, cells were washed with HBSS and WST-1 (Abcam, Cambridge, United Kingdom)

was added to the plate and incubated for 30 min at 37 °C and 5% $CO_2$ to measure the cell viability. Absorbance was read at 440 nm to assess cell viability. Mitochondrial ROS production was normalized to cell viability.

## Statistics

Agilent array data were processed and normalized using LOWESS in Gene Spring GX7.3 (Agilent Technologies). The LOWESS normalized data were further analyzed using significance analysis of microarrays (SAM) for two-class unpaired data to detect expression differences based on q-values (<5%; *Tusher et al., 2001*). The input for SAM was gene expression measurements from a set of microarray experiments, as well as a response variable from each experiment. SAM used simple median centering of the arrays is an unbiased statistical technique for finding significant genes in a set of microarray experiments. SAM uses repeated permutations of the data to determine whether the expression of each gene is significantly related to the response. Significance levels were set at a q-value of 5%. We used a cutoff of the absolute value of $\log_2$ red channel/green channel >0.5. Data were analyzed using GraphPad Prism software. p-Values were calculated using Student's t-test or One-way ANOVA and Tukey' test. Results were expressed as mean ± SD (number of experiments) and considered to be statistically significant when $p < 0.05$.

## Materials availability

We did not use human or animal models, but we have used human cells for this study. Cystinosis RPTE and fibroblast cells are gifts from Dr. Gahl and Dr. Racusen. In addition, the cell lines are commercially purchased. CRISPR-edited renal cell line is available in Sarwal Lab. Please email: Minnie.sarwal@ucsf.edu or Swastika.sur@ucsf.edu. The study was controlled by institutional review board approvals from the National Institute of Health, Stanford University and the Regents, University of California.

## Acknowledgements

We thank Dr. William Gahl and Dr. Lorraine Racusen for the generous gift of cystinosis RPTE and fibroblast cells and Reena Zalpuri at the University of California Berkeley Electron Microscope Laboratory for advice and assistance in electron microscopy sample preparation and data collection. In addition, we would also like to thank Jon Gangoiti at the UCSD Biochemical Genetics Laboratory for advice on sample preparations and performing HPLC-MS/MS method to measure intracellular cystine levels. This work was supported by grants from the Health Research Board, Ireland and The Cystinosis Foundation, Ireland.

## Additional information

### Funding

| Funder | Grant reference number | Author |
| --- | --- | --- |
| Health Research Board | | Minnie M Sarwal |
| Cystinosis Foundation | | Minnie M Sarwal |

The funders had no role in study design, data collection and interpretation, or the decision to submit the work for publication.

### Author contributions

Swastika Sur, Conceptualization, Data curation, Formal analysis, Supervision, Validation, Investigation, Visualization, Methodology, Writing – original draft, Project administration, Writing – review and editing; Maggie Kerwin, Formal analysis, Validation, Visualization, Methodology, Writing – review and editing; Silvia Pineda, Software, Formal analysis, Visualization; Poonam Sansanwal, Methodology; Tara K Sigdel, Writing – review and editing; Marina Sirota, Software, Writing – review and editing; Minnie M Sarwal, Resources, Supervision, Funding acquisition, Project administration, Writing – review and editing

## Author ORCIDs

Swastika Sur (iD) https://orcid.org/0000-0001-9536-981X
Maggie Kerwin (iD) https://orcid.org/0000-0002-6792-342X
Minnie M Sarwal (iD) https://orcid.org/0000-0003-1212-3959

Reviewer #3 (Public review): https://doi.org/10.7554/eLife.94169.3.sa1
Author response https://doi.org/10.7554/eLife.94169.3.sa2

## Additional files

### Supplementary files

MDAR checklist

### Data availability

The transcriptomic data was submitted to GEO. The GEO Accession number is GSE190500.

The following dataset was generated:

| Author(s) | Year | Dataset title | Dataset URL | Database and Identifier |
|---|---|---|---|---|
| Sur S, Kerwin M, Pineda S, Sansanwal P, Sigdel TK, Sirota M, Sarwal MM | 2023 | Human Renal Cells and Skin Fibroblast from Individuals with Nephropathic Cystinosis and their Controls | https://www.ncbi.nlm.nih.gov/geo/query/acc.cgi?acc=GSE190500 | NCBI Gene Expression Omnibus, GSE190500 |

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
