## [Editor Report · eLife Assessment]

This **important** study addresses the idea that defective lysosomal clearance might be causal to renal dysfunction in cystinosis. With mostly **solid** data, the authors observe that restoring expression of vATPase subunits and treatment with Astaxanthin ameliorate mitochondrial function in a model of renal epithelial cells, opening opportunities for translational application to humans.

---

## [Referee Report · Reviewer #3 (Public review)]

Summary:

In this manuscript, Sur and colleagues present insights into the potential pathways and mechanisms underlying the pathogenesis of cystinosis - a prototypical lysosomal storage disorder caused by the loss of the cystine transporter cystinosin (CTNS). This deficiency results in early dysfunction of proximal tubule (PT) cells and proximal tubulopathy, which progresses to chronic kidney disease and multisystem complications later in life. The authors utilize patient-derived cell lines and knockout (KO) strategies in immortalized PT cell systems, alongside transcriptomics and pathway enrichment analyses, to demonstrate that the loss of CTNS function reduces V-ATPase subunits (specifically V-ATP6V0A1), impairing autophagy and mitochondrial homeostasis. These findings are consistent with their prior work and follow-up studies conducted in preclinical models (mouse, rat, and zebrafish) of cystinosis and CTNS deficiency.

Importantly, the authors highlight rescue strategies that involve correcting V-ATP6V0A1 expression or modulating redox dyshomeostasis through ATX treatment. These interventions restore cellular homeostasis in patient-derived cells, providing actionable therapeutic targets for patients in need of novel causal therapies.

Strengths:

The implications for health, disease, and therapeutic discovery are considerable, given the central role of autophagy and lysosome-related pathways in regulating critical cellular processes and physiological functions.

Weaknesses:

Despite these promising findings, further experimental research is required to strengthen the study's framework and conclusions. This includes characterizing the physiological properties of the PT cellular systems used, performing appropriate control or sentinel experiments in lysosome function assays, and further delineating disease phenotypes associated with cystinosis. Follow-up investigations into lysosome abnormalities and autophagy dysfunctions are also needed, along with a detailed exploration of the molecular mechanisms underlying the rescue of lysosomal, autophagic, and mitochondrial phenotypes through ATX treatment.

---

## [Author Response]

The following is the authors’ response to the original reviews.

**Public Reviews:**

**Reviewer #1 (Public Review):**
Cystinosis is a rare hereditary disease caused by biallelic loss of the CTNS gene, encoding two cystinosin protein isoforms; the main isoform is expressed in lysosomal membranes where it mediates cystine efflux whereas the minor isoform is expressed at the plasma membrane and in other subcellular organelles. Sur et al proceed from the assumption that the pathways driving the cystinosis phenotype in the kidney might be identified by comparing the transcriptome profiles of normal vs CTNS-mutant proximal tubular cell lines. They argue that key transcriptional disturbances in mutant kidney cells might not be present in non-renal cells such as CTNS-mutant fibroblasts.Using cluster analysis of the transcriptomes, the authors selected a single vacuolar H+ATPase (ATP6VOA1) for further study, asserting that it was the "most significantly downregulated" vacuolar H+ATPase (about 58% of control) among a group of similarly downregulated H+ATPases. They then showed that exogenous ATP6VOA1 improved CTNS(-/-) RPTEC mitochondrial respiratory chain function and decreased autophagosome LC3-II accumulation, characteristic of cystinosis. The authors then treated mutant RPTECs with 3 "antioxidant" drugs, cysteamine, vitamin E, and astaxanthin (ATX). ATX (but not the other two antioxidant drugs) appeared to improve ATP6VOA1 expression, LC3-II accumulation, and mitochondrial membrane potential. Respiratory chain function was not studied. RTPC cystine accumulation was not studied.

In this manuscript, as an initial step, we have studied the first step in respiratory chain function by performing the Seahorse Mito Stress Test to demonstrate that the genetic manipulation (knocking out the *CTNS* gene and plasmid-mediated expression correction of *ATP6V0A1*) impacts mitochondrial energetics. We did not investigate the respirometry-based assays that can identify locations of electron transport deficiency, which we plan to address in a follow-up paper.

We would like to draw attention to Figure 3D, where cystine accumulation has been studied. This figure demonstrates an increased intracellular accumulation of cystine.

The major strengths of this manuscript reside in its two primary findings.(1) Plasmid expression of exogenous ATP6VOA1 improves mitochondrial integrity and reduces aberrant autophagosome accumulation.(2) Astaxanthin partially restores suboptimal endogenous ATP6VOA1 expression.Taken together, these observations suggest that astaxanthin might constitute a novel therapeutic strategy to ameliorate defective mitochondrial function and lysosomal clearance of autophagosomes in the cystinotic kidney. This might act synergistically with the current therapy (oral cysteamine) which facilitates defective cystine efflux from the lysosome.There are, however, several weaknesses in the manuscript.(1) The reductive approach that led from transcriptional profiling to focus on ATP6VOA1 is not transparent and weakens the argument that potential therapies should focus on correction of this one molecule vs the other H+ ATPase transcripts that were equally reduced - or transcripts among the 1925 belonging to at least 11 pathways disturbed in mutant RPTECs.

The transcriptional profiling studies on ATP6V0A1 have been fully discussed and publicly shared. Table 2 lists the v-ATPase transcripts that are significantly downregulated in cystinosis RPTECs. We have also clarified and justified the choice of further studies on ATP6V0A1, where we state the following: "The most significantly perturbed member of the V-ATPase gene family found to be downregulated in cystinosis RPTECs is ATP6V0A1 (Table 2). Therefore, further attention was focused on characterizing the role of this particular gene in a human in vitro model of cystinosis."

(2) A precise description of primary results is missing -- the Results section is preceded by or mixed with extensive speculation. This makes it difficult to dissect valid conclusions from those derived from less informative experiments (eg data on CDME loading, data on whole-cell pH instead of lysosomal pH, etc).

We appreciate the reviewer highlighting areas for further improving the manuscript's readership. In our resubmission, we have revised the results section to provide a more precise description of the primary findings and restrict the inferences to the discussion section only.

(3) Data on experimental approaches that turned out to be uninformative (eg CDME loading, or data on whole=cell pH assessment with BCECF).

We have provided data whether it was informative or uninformative. Though lysosome-specific pH measurement would be important to measure, it was not possible to do it in our cells as they were very sick and the assay did not work. Hence we provide data on pH assessment with BCECF, which measures overall cytoplasmic and organelle pH, which is also informative for whole cell pH that is an overall pH of organelle pH and cytoplasmic pH.

(4) The rationale for the study of ATX is unclear and the mechanism by which it improves mitochondrial integrity and autophagosome accumulation is not explored (but does not appear to depend on its anti-oxidant properties).

We have provided rationale for the study of ATX; provided in the introduction and result section, where we mentioned the following: “correction of ATP6V0A1 in CTNS-/- RPTECs and treatment with antioxidants specifically, astaxanthin (ATX) increased the production of cellular ATP6V0A1, identified from a custom FDA-drug database generated by our group, partially rescued the nephropathic RPTEC phenotype. ATX is a xanthophyll carotenoid occurring in a wide variety of organisms. ATX is reported to have the highest known antioxidant activity and has proven to have various anti-inflammatory, anti-tumoral, immunomodulatory, anti-cancer, and cytoprotective activities both in vivo and in vitro_”._

We are still investigating the mechanism by which ATX improves mitochondrial integrity, and this will be the focus of a follow-on manuscript.

(5) Thoughtful discussion on the lack of effect of ATP6VOA1 correction on cystine efflux from the lysosome is warranted, since this is presumably sensitive to intralysosomal pH.

In the revised manuscript, we have included a detailed discussion on the plausible reasons why ATP6V0A1 correction has no effect on cysteine efflux from the lysosome. We have now added to the Discussion – “However, correcting ATP6V0A1 had no effect on cellular cystine levels, likely because cystinosin is known to have multiple roles beyond cystine transport Cystinosin is demonstrated to be crucial for activating mTORC1 signaling by directly interacting with v-ATPases and other mTORC1 activators. Cystine depletion using cysteamine does not affect mTORC1 signaling. Our data, along with these observations, further supports that cystinosin has multiple functions and that its cystine transport activity is not mediated by ATP6V0A1.”

(6) Comparisons between RPTECs and fibroblasts cannot take into account the effects of immortalization on cell phenotype (not performed in fibroblasts).

The purpose of examining different tissue sources of primary cells in nephropathic cystinosis was to assess if any of the changes in these cells were tissue source specific. We used primary cells isolated from patients with nephropathic cystinosis—RPTECs from patients' urine and fibroblasts from patients' skin—these cells are not immortalized and can therefore be compared. This is noted in the results section - “Specific transcriptional signatures are observed in cystinotic skin-fibroblasts and RPTECs obtained from the same individual with cystinosis versus their healthy counterparts”.

We next utilized the immortalized RPTEC cell line to create CRISPR-mediated *CTNS* knockout RPTECs as a resource for studying the pathophysiology of cystinosis. These cells were not compared to the primary fibroblasts.

(7) This work will be of interest to the research community but is self-described as a pilot study. It remains to be clarified whether transient transfection of RPTECs with other H+ATPases could achieve results comparable to ATP6VOA1. Some insight into the mechanism by which ATX exerts its effects on RPTECs is needed to understand its potential for the treatment of cystinosis.

In future studies we will further investigate the effect of ATX on RPTECs for treatment of cystinosis- this will require the conduct of Phase 1 and Phase 2 clinical studies which are beyond the scope of this current manuscript.

**Reviewer #2 (Public Review):**
Sur and colleagues investigate the role of ATP6V0A1 in mitochondrial function in cystinotic proximal tubule cells. They propose that loss of cystinosin downregulates ATP6V0A1 resulting in acidic lysosomal pH loss, and adversely modulates mitochondrial function and lifespan in cystinotic RPTECs. They further investigate the use of a novel therapeutic Astaxanthin (ATX) to upregulate ATP6V0A1 that may improve mitochondrial function in cystinotic proximal tubules.The new information regarding the specific proximal tubular injuries in cystinosis identifies potential molecular targets for treatment. As such, the authors are advancing the field in an experimental model for potential translational application to humans.
**Recommendations for the authors:**

**Reviewer #1 (Recommendations For The Authors):**
(1) There is a lack of care with precise wording and punctuation, which negatively affects the text. Importantly, the manuscript lacks a clear description of experimental Results. This section begins with speculation, then wanders through experimentation that didn't work (could be deleted). Figure 1A and lines 94-102 could be deleted. Data from CDME loading was found to be a "poor surrogate" for cystinosis and could be deleted from the manuscript or mentioned as a minor point in the discussion. The number of individual patient cell lines used for experimentation is unclear - 8 patients are mentioned on line 109, Figure 2B shows 6 normal fibroblasts, 3 CDME-loaded fibroblasts, and an indeterminate number of normal vs CDME-loaded cells (both colored red). Cluster analysis refers to two large gene clusters - data supporting this key conclusion is not shown. It is unclear why ATP6VOA1 was selected as the most significantly reduced H+ATPase from Table II. Thus, the focus on this particular gene appears to be largely "a hunch".

In this study, we aim to establish a new concept by using multiple cell types and various assays tailored to each affected organelle, which might be confusing. Therefore, we believe Figure 1a provides a roadmap and helps clarify what to expect from this paper.

This study was started a decade back, when CDME-mediated lysosomal loading was regularly used as a surrogate in vitro model to study cystinosis tissue injury. That was the reason to include CDME in the study design. Since we already had the CDME-treated data and in this article we are talking about another superior in vitro cystinosis model, we would like to include it.

In the Result and Methods section, we mentioned “8 patients” with nephropathic cystinosis from whom we collected the RPTECs and Fibroblasts. These cystinotic cells are shown in blue and purple dots, respectively in figure 2B. Normal RPTEC and fibroblast cells were purchased from company and these cells were then treated with CDME to artificially load lysosomes with cystine. Details on the cell types and its procurement can be found in the Methods section under “Study design and Samples”. Normal and CDME-loaded RPTECs are shown in red and orange dots, whereas normal and CDME-loaded fibroblasts are shown in green and yellow dots, respectively in figure 2B.

We removed this figure from the manuscript because the data is already detailed in Tables 1 and 2. As a sub-figure, the string pathway analysis output was illegible and did not add any new information. However, for your reference, we have now provided this data below.

**Author response image 1. sa2fig1:** STRIG pathway analysis using the microarray transcriptomic data from normal vs. cystinotic RPTECs. Ysing K-mean clustering on the genes in these significantly enriched pathways, we identified 2 distinct clusters, red and green nodes. Red nodes are enriched in nucleus-encoded mitochondrial genes and v-ATPases family, which are crucial for lysosomes and kidney tubular acid secretion. ATP6VOA1, the topmost v-ATPase in our cystinotic transcriptome dataset is highlighted in cyan. Green nodes are enriched in genes needed for DNA replication.

(2) It was decided to use transcriptional profiling of CTNS mutant vs wildtype renal proximal tubular cells (RPTECs) as a way to uncover defective secondary molecular pathways that might be upstream drivers of the cystinosis phenotype. Since the kidneys are the first organs to deteriorate in cystinosis, it is postulated that transcriptome differences might be more obvious in kidney cells than in non-renal tissues, such as fibroblasts. A potential pitfall is that the RPTECs were transformed cell lines whereas fibroblasts were not.

Transcriptional profiling was done on primary cells isolated from patients with nephropathic cystinosis—RPTECs from patients' urine and fibroblasts from patients' skin—these cells are not immortalized and can therefore be compared. This is noted in the results section - “Specific transcriptional signatures are observed in cystinotic skin-fibroblasts and RPTECs obtained from the same individual with cystinosis versus their healthy counterparts”.

We utilized the immortalized RPTEC cell line to create CRISPR-mediated *CTNS* knockout RPTECs as a resource for studying the pathophysiology of cystinosis. These cells were not compared to the primary fibroblasts.

(3) The authors wanted to study intralysosomal pH but could not, so used a pH-sensitive dye that reflects whole cell pH. It would be incorrect to take this measurement as support for their hypothesis that intralysosomal pH is increased. Since these experiments cannot be interpreted, they should be deleted from the manuscript.

We have now corrected the term to "intracellular pH." Although measuring lysosome-specific pH would be important, it was not feasible in our cells as knocking out cystinosin gene made them fragile, making the assay ineffective. Therefore, we provide data on pH assessment using BCECF, which measures the overall pH of the cytoplasm and organelles. This information is still valuable for understanding the whole cell pH, encompassing both organelle and cytoplasmic pH. We have mentioned this as one of our limitations in the Discussion section.

(4) The choice of ATX as a potential therapy is puzzling. Its antioxidant properties seem to be irrelevant since two other antioxidants had no effect. The mechanism by which it appears to correct some aspects of the cystinosis phenotype remains unknown and this should be pointed out. A key experiment to assess whether ATX reduces lysosomal cystine accumulation is missing. While the impact of ATX on cystinosis is interesting, the mechanism is unexplored.

A detailed study on the mechanism by which ATX corrects certain aspects of the cystinosis phenotype is currently underway and will be presented in a follow-up paper. We have measured the effect of ATX and cysteamine, both individually and combined, on cystine accumulation using HPLC, as shown in the figure below. Our results indicate a significant increase in cystine levels with ATX treatment alone, while the combined ATX and cysteamine treatment significantly reduced cystine accumulation to the normal level. This suggests that ATX addresses specific aspects of the cystinosis phenotype through a different mechanism, not by reducing the accumulated cystine levels. When co-administered with cysteamine, they have the potential to complement each other's shortcomings. We believe that the increase in cystine with ATX alone may be due to interactions between ATX's ketone or hydroxyl groups and cystine's amine or carboxylic groups. Further research on this interaction is ongoing.

We have now added to the Discussion – “We noticed a significant increase in cystine levels with ATX treatment alone (data not shown in the manuscript), while the combined ATX and cysteamine treatment significantly reduced cystine accumulation to the normal level. This may suggest that when co-administered with cysteamine, they have the potential to complement each other's shortcomings. We believe that the increase in cystine with ATX alone could be due to interactions between ATX's ketone or hydroxyl groups and cystine's amine or carboxylic groups. Further research on this interaction is ongoing.”

(5) The effects of exogenous ATP6VOA1 are interesting but had no effect on lysosomal cystine efflux, a hallmark of the cystinosis cellular phenotype. A discussion of this issue would be important.

In the revised manuscript, we have included a detailed discussion on the plausible reasons why ATP6V0A1 correction has no effect on cysteine efflux from the lysosome. We have added to the Discussion – “However, correcting ATP6V0A1 had no effect on cellular cystine levels (Figure 7C), likely because cystinosin is known to have multiple roles beyond cystine transport. Cystinosin is demonstrated to be crucial for activating mTORC1 signaling by directly interacting with v-ATPases and other mTORC1 activators. Cystine depletion using cysteamine does not affect mTORC1 signaling (47). Our data, along with these observations, further supports that cystinosin has multiple functions and that its cystine transport activity is not mediated by ATP6V0A1.”

(6) The arguments on lines 260-273 are not comprehensible. The authors confirm that RPTC LC3-II levels are increased, a marker of active processing of autophagosome cargo, prior to delivery to lysosomes. Discussion of balfilomycin (not used), mTORC activity, and endocytosis are not directly relevant and wander from interpretation of the LC3-II observation. One possibility is that the 50% decrease in ATP6VOA1 transcript is sufficient to slow the transfer of LC3-II-tagged cargo from autophagosome to lysosome - however, it would be important to offer a plausible explanation for why decreased ATP6VOA1 expression alone does not appear to be the key limitation on lysosomal cystine efflux.

We have now rephrased our explanation in the Discussion section – “Cystinotic cells are known to have an increased autophagy or reduced autophagosome turnover rate. Autophagic flux in a cell is typically assessed by examining the accumulation of the autophagosome or autophagy-lysosome marker LC3B-II. This accumulation can be artificially induced using bafilomycin, which targets the V-ATPase, thereby inhibiting lysosomal acidification and degradation of its contents. Taken together, the observed innate increase in LC3B-II in cystinotic RPTECs (Figure 5A) without bafilomycin treatment suggests dysfunctional lysosomal acidification and thus could be linked to inhibited v-ATPase activity”.